# Self-guided Robust Graph Structure Refinement

## ABSTRACT

Recent studies have revealed that GNNs are vulnerable to adversarial attacks. To defend against such attacks, robust graph structure refinement (GSR) methods aim at minimizing the effect of adversarial edges based on node features, graph structure, or external information. However, we have discovered that existing GSR methods are limited by narrow assumptions, such as assuming clean node features, moderate structural attacks, and the availability of external clean graphs, resulting in the restricted applicability in real-world scenarios. In this paper, we propose a **s**elf-**g**uided GSR framework (SG-GSR), which utilizes a clean sub-graph found within the given attacked graph itself. Furthermore, we propose a novel graph augmentation and a group-training strategy to handle the two technical challenges in the clean sub-graph extraction: 1) loss of structural information, and 2) imbalanced node degree distribution. Extensive experiments demonstrate the effectiveness of SG-GSR under various scenarios including non-targeted attacks, targeted attacks, feature attacks, e-commerce fraud, and noisy node labels. Our code is available at https://anonymous.4open.science/r/torch-SG-GSR-97CC.

## 1 INTRODUCTION

A graph is a widely-used data structure in many domains. Graph neural networks (GNNs) have shown success in learning node representations in graphs [13, 14, 32] and have been applied to various tasks, such as node classification [42], link prediction [43], and recommender systems [36]. Despite the advancement of GNNs, recent research has found that GNNs are vulnerable to adversarial attacks [5, 38, 41, 50, 51]. *Adversarial attack on a graph* aims at injecting small and imperceptible changes to the graph structure and node features that easily fool a GNN to yield wrong predictions. In other words, even slight changes in the graph (e.g., adding a few edges [51] or injecting noise to the node features [22]) can significantly deteriorate the predictive power of GNN models, which raises concerns about their use in various real-world applications. For example, given a product co-reivew graph in a real-world e-commerce platform, attackers would write fake product reviews on arbitrary products, aiming to manipulate the structure of the product graph and the node (i.e., product) features, thereby fooling the models into predicting the wrong co-review links or product categories.

Graph structure refinement (GSR) methods [4, 8, 12, 17, 20, 30, 35] have been recently demonstrated to improve the robustness of GNNs by minimizing the impact of adversarial edges during message passing. These methods can be categorized based on the type

*Conference'17, July 2017, Washington, DC, USA*
© 2018 Association for Computing Machinery.
ACM ISBN 978-x-xxxx-xxxx-x/YY/MM. . . $15.00
https://doi.org/XXXXXXX.XXXXXXX

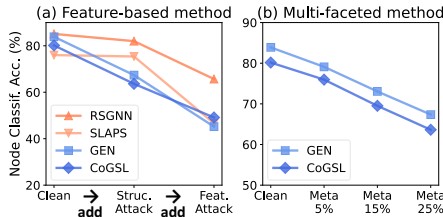

**Figure 1: Performance of (a) existing feature-based and multi-faceted GSR methods over structure (Meta 25%) and feature attacks (Random Gaussian noise 50%), (b) existing multi-faceted methods under different perturbation ratios. Cora is used. Meta:** *metattack* **[51].**

of information used to refine the graph structure. The first line of research utilizes the node feature information [4, 8, 12], whose main idea is to encourage the nodes with similar features to be connected, i.e., feature smoothness assumption [12]. However, these approaches cannot be applied when the node features are not available, and more importantly, their performance drops significantly when the node features are noisy or attacked [20, 29]. Fig. 1(a) demonstrates that the performance of two recent feature-based GSR methods, RSGNN [4] and SLAPS [8], drops significantly when the node features are noisy or attacked (i.e., add Feat. Attack). In other words, relying heavily on the node features unavoidably results in a performance drop when the node features are noisy or attacked.

To address the limitation of feature-based GSR methods, another line of research utilizes the multi-faceted information [20, 29, 35], i.e., both the node features and the graph structural information. Their main idea is to exploit the high-order structural similarity in addition to the node feature similarity to refine the attacked graph structure. However, we argue that additionally utilizing the graph structural information is helpful only when the amount of the attack on the given graph is moderate. Fig. 1(b) demonstrates that the performance of two recent multi-faceted methods, GEN [35] and CoGSL [20], drops notably as the perturbation ratio of structure attack increases (i.e., from Meta 5% to Meta 25%). In other words, when the given graph is heavily attacked, utilizing the graph structural information is sub-optimal as the structure of the given graph itself contains a lot of adversarial edges. A possible solution to this issue would be to replace the attacked graph structure with a clean proxy structure. PA-GNN [30] employs external clean graphs obtained from similar domains to which the target attacked graph belongs as the proxy structure. However, we emphasize that it is not practical, hence not applicable in reality due to its strong assumption on the existence of external clean graphs. In summary, existing GSR methods are limited by narrow assumptions, such as assuming clean node features, moderate structural attacks, and the availability of external clean graphs, resulting in the restricted applicability in real-world scenarios.

To mitigate the aforementioned problems, we propose a **s**elf-**g**uided GSR framework (SG-GSR), which is a multi-faceted GSR method that utilizes a clean proxy structure in addition to the node feature information. The proposed method consists of three steps: (Step 1) extracting a confidently clean *sub-graph* from the target

attacked graph , (Step 2) training a robust GSR module based on the sub-graph that is considered as the clean proxy structure, and (Step 3) using the knowledge obtained from the clean proxy structure to refine the target attacked graph and learn a robust node classifier.

However, there exist two technical challenges when extracting the clean sub-graph from an attacked graph. The first challenge is the *loss of structural information*. When extracting a clean sub-graph by removing edges that are predicted to be adversarial, we observe that a considerable amount of the removed edges are indeed clean edges, thereby limiting the robustness of the GSR module. The second challenge is the *imbalanced node degree distribution of the clean sub-graph*, which inhibits the generalization ability of the GSR module to low-degree nodes. More precisely, since the average number of edges incident to a low-degree node in the imbalanced sub-graph is greatly smaller than that of a high-degree node, the GSR module trained on the imbalanced sub-graph would be biased towards high-degree nodes. Note that even though the ability is of great importance to the overall performance since a majority of nodes are of low-degree in real-world graphs, there are few existing works dealing with the low-degree nodes in the context of robust GSR.

To further handle the above two challenges of the clean sub-graph extraction, we propose **1)** a novel graph augmentation strategy to supplement the loss of structural information of the extracted sub-graph, thereby enhancing the robustness of the GSR module to attacks in the target graph, and **2)** a group-training strategy that independently trains the GSR module for each node group, where the node groups are constructed based on the node degree distribution in a balanced manner, thereby enhancing the generalization ability of the GSR module to low-degree nodes.

In summary, the main contributions of this paper are three-fold:

- We discover the narrow assumptions of existing GSR methods limit their applicability in the real-world (Fig. 1), and present a novel self-guided GSR framework, called SG-GSR, that achieves adversarial robustness by extracting the clean sub-graph while addressing its two technical challenges: 1) loss of structural information and 2) imbalanced node degree distribution.
- SG-GSR outperforms state-of-the-art baselines in node classification, and we show its effectiveness under various scenarios including non-targeted attacks, targeted attacks, feature attacks, e-commerce fraud, and noisy node labels.
- We introduce novel graph benchmark datasets that simulate real-world fraudsters' attacks on e-commerce systems, as an alternative to artificially attacked graph datasets, which is expected to foster practical research in adversarial attacks on GNNs.

## 2 RELATED WORKS

### 2.1 Robust GNNs

Robust GNN methods include approaches based on graph structure refinement [4, 12], adversarial training [15], Gaussian distribution-based node representation learning [47], new message passing scheme [16, 22, 23], leveraging low-rank components of the graph [7], and etc. Among these methods, one representative approach is graph structure refinement (GSR), which aims to learn a better graph structure from a given graph, and it has recently been adopted to mitigate the impact of adversarial edges in attacked graphs. In the following, we briefly introduce existing GSR methods.

**Feature-based GSR.** The first line of research utilizes node features under the feature smoothness assumption [4, 8, 12]. ProGNN [12] refines the attacked graph structure by satisfying numerous real-world graph properties, e.g., feature-smoothness, sparsity, and low-rankness. SLAPS [8] trains an MLP encoder to produce a new graph structure where edges connect nodes with similar embeddings. RSGNN [4] uses an MLP encoder and a regularizer that encourages similar nodes to be close in the representation space. However, we demonstrate that relying heavily on the node features unavoidably results in a performance drop when the node features are noisy or attacked as shown in Fig. 1(a).

**Multi-faceted GSR.** To handle the weakness of feature-based approaches, another line of research leverages multi-faceted information that considers the structural information in addition to the node features. GEN [35] estimates a new graph structure via Bayesian inference from node features and high-order neighborhood information. CoGSL [20] aims to learn an optimal graph structure in a principled way from the node features and structural information. However, we demonstrate that using additional structural information is sub-optimal when the graph structure is heavily attacked as shown in Fig. 1(b). To address this issue, PA-GNN transfers knowledge from clean external graphs to improve inference on attacked graphs. However, it assumes the existence of external clean graphs, which is not practical and realistic as real-world graphs contain inherent noise. Moreover, STABLE [17] is a contrastive-learning method that maximizes mutual information between representations from the graph views generated by randomly removing easily detectable adversarial edges. However, a significant number of the removed edges are indeed clean edges, which causes a severe loss of vital structural information, thereby limiting the robustness of GSR (Refer to Appendix C.1 for more details).

Different from the aforementioned methods, we aim to refine the attacked graph structure based on multi-faceted information by utilizing the clean sub-graph instead of the attacked structure. In doing so, we handle the two technical challenges of extracting the sub-graph that hinder the robustness of GSR, i.e., loss of structural information and imbalanced node degree distribution.

### 2.2 Imbalanced Learning on Node Degree

The node degrees of many real-world graphs follow a power-law (i.e., a majority of nodes are of low-degree). However, GNNs heavily rely on structural information for their performance, which can result in underrepresentation of low-degree nodes [24, 25]. To tackle the issues regarding low-degree nodes, Meta-tail2vec [25] and Tail-GNN [24] propose ways to refine the representation of low-degree nodes by transferring information from high-degree nodes to low-degree nodes. Despite their effectiveness, they do not consider the problem in the context of adversarial attacks, but they simply assume that the given graph is clean. Although recent studies [17, 51] demonstrate that low-degree nodes are more vulnerable to adversarial attacks than high-degree nodes, existing GSR methods do not pay enough attention to low-degree nodes. One straightforward solution would be to mainly use the node features that are independent of the node degree, such as in SLAPS [8] and RSGNN [4]. However, as shown in Fig. 1(a), their performance deteriorates when the node features are noisy or attacked. In this work, we propose a novel GSR method that directly focuses on enhancing the robustness of GSR with respect to low-degree nodes by balancing the node degree

distribution. Moreover, by utilizing the clean proxy structure (i.e., clean sub-graph) in addition to the node features, our proposed method is more robust even when the node features are attacked.

## 3 PROBLEM STATEMENT

We use $\mathcal{G} = \langle \mathcal{V}, \mathcal{E}, \mathbf{X} \rangle$ to denote an attacked graph, where $\mathcal{V} = \{v_1, ..., v_N\}$ is the set of nodes, $\mathcal{E} \in \mathcal{V} \times \mathcal{V}$ is the set of edges, and $\mathbf{X} \in \mathbb{R}^{N \times F}$ is the node feature matrix, where $N$ is the number of nodes, and $F$ is the number of features for each node. We use $\mathbf{A} \in \mathbb{R}^{N \times N}$ to denote the adjacency matrix where $\mathbf{A}_{ij} = 1$ if an edge exists between nodes $i$ and $j$, otherwise $\mathbf{A}_{ij} = 0$. We assume the semi-supervised setting, where only a portion of nodes are labeled. The class label of a labeled node $i$ is defined as $\mathbf{Y}_i \in \{0, 1\}^C$, where $C$ indicates the number of classes. Our goal is to learn a robust node classifier based on the refined graph structure.

## 4 PROPOSED METHOD

We propose a **s**elf-**g**uided GSR framework (SG-GSR), whose main idea is to train a robust GSR module (**Sec 4.1**) based on a confidently clean sub-graph extracted from the given attacked graph (**Sec 4.2**). We further explore and deal with the two technical challenges of extracting a clean sub-graph, i.e., loss of structural information (**Sec 4.3.1**) and imbalanced node degree distribution (**Sec 4.3.2**), by introducing two strategies, a graph augmentation (**Sec 4.4.1**) and group-training (**Sec 4.4.2**), respectively. Finally, we use the knowledge obtained from training the GSR module with two strategies in order to refine the target attacked graph and learn a robust node classifier. (**Sec 4.5**). Appendix A shows the overall architecture of SG-GSR.

## 4.1 Graph Structure Refinement (GSR) Module

We adopt SuperGAT [13] as the backbone network for refining the attacked graph $\mathcal{G}$. In SuperGAT with $L$ layers, the model transforms the representations of node $i$ for layer $l$, i.e., $\mathbf{h}_i^l \in \mathbb{R}^{F^l}$, using a weight matrix $\mathbf{W}^{l+1} \in \mathbb{R}^{F^{l+1} \times F^l}$, and the updated node representation $\mathbf{h}_i^{l+1} \in \mathbb{R}^{F^{l+1}}$ is obtained by linearly combining the representations of node $i$ and its first-order neighbors $j \in \mathcal{N}_i$ using attention coefficients, i.e., $\alpha_{ij}^{l+1}$, which is followed by non-linear activation $\rho$ as: $\mathbf{h}_i^{l+1} = \rho \left( \sum_{j \in \mathcal{N}_i \cup \{i\}} \alpha_{ij}^{l+1} \mathbf{W}^{l+1} \mathbf{h}_j^l \right)$, where $\alpha_{ij}^{l+1} =$ softmax$_{j \in \mathcal{N}_i \cup \{i\}} (\rho(e_{ij}^{l+1}))$. Note that $F^L$ is set to $C$, which is the number of classes. Among various ways to compute $e_{ij}^{l+1}$, we adopt the dot-product attention [31]: $e_{ij}^{l+1} = [(\mathbf{W}^{l+1} \mathbf{h}_i^l)^\top \cdot \mathbf{W}^{l+1} \mathbf{h}_j^l] / \sqrt{F^{l+1}}$. We pass the output of the final layer, i.e., $\mathbf{h}_i^L$, through a softmax function to generate the prediction of node labels, i.e., $\hat{Y}_i$, which is then used to compute the cross-entropy loss as follows:

$$L_\mathcal{V} = - \sum_{i \in \mathcal{V}^L} \sum_{c=1}^{C} \mathbf{Y}_{ic} \log \hat{Y}_{ic} \quad (1)$$

where $\mathcal{V}^L$ indicates the labeled node set. In each layer $l$, to learn $\mathbf{W}^l$ that makes $e_{ij}^l$ large for clean edges, and small for adversarial edges, we optimize the following link prediction loss $L_\mathcal{E}^l$ in addition to the cross-entropy loss $L_\mathcal{V}$:

$$L_\mathcal{E}^l = - \left( \frac{1}{|\mathcal{E}|} \sum_{(i,j) \in \mathcal{E}} \cdot \log \phi_{ij}^l + \frac{1}{|\mathcal{E}^-|} \sum_{(i,j) \in \mathcal{E}^-} \cdot \log \left( 1 - \phi_{ij}^l \right) \right) \quad (2)$$

where $\phi_{ij}^l = \sigma(e_{ij}^l)$ indicates the probability that there exists an edge between two nodes $i$ and $j$, which is computed in each layer $l$, and $\sigma$ is the sigmoid function. $\mathcal{E}$ is the set of observed edges, which are considered as clean edges (i.e., positive samples), and $\mathcal{E}^-$ is the set of unobserved edges considered as adversarial edges (i.e., negative samples), which is sampled uniformly at random from the complement set of $\mathcal{E}$. By minimizing $L_\mathcal{E}^l$, we expect that a large $e_{ij}^l$ is assigned to clean edges, whereas a small $e_{ij}^l$ is assigned to adversarial edges, thereby minimizing the effect of adversarial edges. However, since $\mathcal{E}$ may contain unknown adversarial edges due to structural attacks, the positive samples in Eqn. 2 may contain false positives, which leads to a sub-optimal solution under structural attacks. Appendix B.1 clearly shows the negative impact of the false positive edges.

## 4.2 Extraction of Clean Sub-graph (Phase 1)

To alleviate false positive edges , we propose a clean sub-graph extraction method that obtains a clean proxy structure from the target attacked graph, which consists of the following two steps:

(1) **Similarity computation:** We compute the *structural proximity* $S_{ij}^{\text{sp}}$ and node *feature similarity* $S_{ij}^{\text{fs}}$ for all existing edges $(i, j) \in \mathcal{E}$. To compute $S_{ij}^{\text{sp}}$, we use node2vec [9] pretrained node embeddings $\mathbf{H}^{\text{sp}} \in \mathbb{R}^{N \times D^{\text{sp}}}$, which captures the structural information. To compute $S_{ij}^{\text{fs}}$, we use the node feature matrix $\mathbf{X}$, and cosine similarity.

(2) **Sub-graph extraction:** Having computed $S_{ij}^{\text{sp}}$ and $S_{ij}^{\text{fs}}$ for all existing edges $(i, j) \in \mathcal{E}$, we extract the edges with high structural proximity, i.e., $\tilde{\mathcal{E}}^{\text{sp}}$, and the edges with high feature similarity, i.e., $\tilde{\mathcal{E}}^{\text{fs}}$, from the target attacked graph $\mathcal{E}$, where $|\tilde{\mathcal{E}}^*| = \lfloor |\mathcal{E}| \cdot \lambda_* \rfloor$ and $\lambda_* \in [0, 1]$, where $* \in \{\text{sp,fs}\}$. For example, $\lambda_{\text{sp}} = 0.2$ means edges whose $S_{ij}^{\text{sp}}$ value is among top-20% are extracted. Note that $\lambda_{\text{sp}}$ and $\lambda_{\text{fs}}$ are hyperparameters. Lastly, we obtain a clean sub-graph $\tilde{\mathcal{E}}$ by jointly considering $\tilde{\mathcal{E}}^{\text{sp}}$ and $\tilde{\mathcal{E}}^{\text{fs}}$, i.e., $\tilde{\mathcal{E}} = \tilde{\mathcal{E}}^{\text{sp}} \cap \tilde{\mathcal{E}}^{\text{fs}}$, thereby capturing both the structural proximity and the feature similarity.

It is important to note that we constrain the sub-graph size, i.e., $|\tilde{\mathcal{E}}|$, to prevent it from becoming too small, ensuring that the sub-graph always includes labeled nodes. We argue that considering both feature similarity and structural proximity is significant when node features are noisy or attacked. It is also important to note that Phase 1 is done before the model training, and the time complexity of training node2vec is acceptable as it scales linearly with the number of nodes. The supporting results and detailed explanations can be found in Appendix B.2. In Phase 2, we train our proposed GSR module based on the sub-graph extracted in Phase 1, which however poses two challenges.

## 4.3 Challenges on the Extracted Sub-graph

In this subsection, we analyze the two technical challenges of extracting the clean sub-graph that limit the robustness of the proposed GSR method: 1) loss of structural information (**Section 4.3.1**), and 2) imbalanced node degree distribution (**Section 4.3.2**).

*4.3.1 Loss of Structural Information.* Recall that when extracting the clean sub-graph, we only extract a small fraction of confidently clean edges. In other words, we remove a large number of edges from the graph, and these edges may contain numerous clean edges as well as adversarial edges. In Fig. 2(a), we indeed observe that as

 

the ratio of extracted edges gets smaller, i.e., as the ratio of removed edges gets larger, most of the extracted edges are clean (blue line), but at the same time the remaining edges include a lot of clean edges as well (orange line), which incurs the loss of vital structural information. We argue that the limited structural information hinders the predictive power of GNNs on node classification [21, 44] and moreover, restricts the generalization ability of the link predictor in the GSR module (Eqn. 2). As a result, in Fig. 2(b), we observe that although the extracted sub-graph is clean enough (e.g., clean rate is around 0.95 when $|\tilde{\mathcal{E}}|/|\mathcal{E}| = 0.4$), the node classification accuracy is far lower than the clean case, which implies that a naive adoption of the GSR module is not sufficient. Hence, it is crucial to supplement the extracted sub-graph with additional structural information.

*4.3.2 Imbalanced Node Degree Distribution.* We identify another challenge, i.e., imbalanced node degree distribution of the clean sub-graph, that further deteriorates the generalization ability of the link predictor in the GSR module to low-degree nodes. That is, since the average number of edges incident to a low-degree node in the imbalanced sub-graph is greatly smaller than that of a high-degree node, high-degree nodes would dominate the edge set of sub-graph, i.e., $\tilde{\mathcal{E}}$, thereby hindering the generalization ability of the link predictor trained using Eqn. 2 to other nodes (i.e., low-degree nodes). In Fig. 2(c), while both the original graph and the extracted sub-graph are imbalanced, we find that the sub-graph is more severely imbalanced. In Fig. 2(d), we clearly see that when a node is connected to adversarial edges, the accuracy drop in terms of node classification of low-degree nodes compared with the clean case is larger than that of high-degree nodes, which implies that the imbalanced degree distribution of the sub-graph leads to the poor generalization of link predictors to low-degree nodes. This challenge is crucial in many real-world applications since a majority of nodes are of low-degree in real-world graphs.

## 4.4 Dealing with the Challenges of Sub-graph Extraction (Phase 2)

In this subsection, we endeavor to tackle the above challenges that hinder the robustness of the proposed GSR method. Based on the analyses in the previous subsection, we propose 1) a novel graph structure augmentation strategy to supplement the loss of structural information (**Sec 4.4.1**), and 2) a group-training strategy to balance the node degree distribution of the sub-graph (**Sec 4.4.2**).

*4.4.1 Graph Augmentation (Phase 2-1).* To address the first challenge, we propose a novel graph structure augmentation strategy that supplements the structural information of the extracted sub-graph. More specifically, we add edges that are considered to be important for predicting node labels, but *currently non-existent in the extracted sub-graph $\tilde{\mathcal{E}}$.* We measure the importance of each edge based on three real-world graph properties, i.e., class homophily, feature smoothness, and structural proximity.

**Property 1: Class homophily.** An edge is considered to be class homophilic if the two end nodes share the same label, and it is well-known that increasing the class homophily ratio yields better prediction of node labels [3, 45]. Hence, our strategy is to find class homophilic edges and inject them into the sub-graph. However, as only a small portion of nodes are labeled under semi-supervised settings, we need to infer the labels of unlabeled nodes to determine class homophilic edges. To this end, we leverage the class prediction

probability matrix for the set of nodes $\tilde{\mathcal{V}}$ in the extracted sub-graph $\tilde{\mathcal{E}}$, i.e., $\mathbf{P} \in \mathbb{R}^{|\tilde{\mathcal{V}}| \times C}$, as the pseudo-label, and compute the distance between all pairs of nodes based on $\mathbf{P}$. Our intuition is that nodes with a small distance in terms of the class prediction are likely to form a class homophilic edge. Specifically, we adopt the Jensen-Shannon Divergence (JSD) as the distance metric to compute the distance between the class prediction probability of two nodes $i$ and $j$ as follows:

$$\text{JSD}_{ij} = \frac{1}{2}\text{KLD}(\mathbf{P}_i||\mathbf{M}_{ij}) + \frac{1}{2}\text{KLD}(\mathbf{P}_j||\mathbf{M}_{ij}), \text{ for all } i, j \in \tilde{\mathcal{V}} \quad (3)$$

where $\text{KLD}(\cdot||\cdot)$ is the KL-divergence, and $\mathbf{M}_{ij} = (\mathbf{P}_i + \mathbf{P}_j)/2$. In short, we consider an edge in $\{(i,j)|i \in \tilde{\mathcal{V}}, j \in \tilde{\mathcal{V}}\}$ to be class homophilic if the JSD value is small, and thus we add edges with small JSD values (i.e., $\tilde{\mathcal{E}}^{\text{JSD}}$) to the sub-graph to satisfy the class homophily property.

**Property 2: Feature smoothness.** Feature smoothness indicates that the neighboring (or adjacent) nodes share similar node features, which is widely used to inject more structural information for improving node classification accuracy [8, 12]. Hence, based on the node feature matrix $\tilde{\mathbf{X}} \in \mathbb{R}^{|\tilde{\mathcal{V}}| \times F}$ of the extracted sub-graph $\tilde{\mathcal{E}}$, we compute the cosine similarity between all node pairs in the sub-graph. Then, we add edges with high node feature similarity (i.e., $\tilde{\mathcal{E}}^{\text{FS}}$) to the sub-graph to satisfy the feature smoothness property.

**Property 3: Structural proximity.** Structural proximity indicates that structurally similar nodes in a graph tend to be adjacent or close to each other [9, 18, 33, 46]. Although conventional metrics such as Jaccard Coefficient, Common Neighbors [19], and Adamic-Adar [2] are widely used to measure the structural proximity between nodes, they mainly focus on the local neighborhood structures, and thus fail to capture high-order structural similarity. Hence, to capture the high-order structural proximity, we use the pre-trained node embeddings $\mathbf{H}^{\text{sp}}$ obtained by node2vec [9]. node2vec is a random-walk based node embedding method that is known to capture the high-order structure proximity, which is already obtained in Phase 1. We compute the cosine similarity between all node pairs in the extracted sub-graph $\tilde{\mathcal{E}}$ and add the edges with the highest structural proximity values (i.e., $\tilde{\mathcal{E}}^{\text{SP}}$) into the sub-graph.

In summary, we perform augmentations on the extracted sub-graph (i.e., $\tilde{\mathcal{E}}$) as: $\tilde{\mathcal{E}}^{\text{aug}} = \tilde{\mathcal{E}} \cup \tilde{\mathcal{E}}^{\text{JSD}} \cup \tilde{\mathcal{E}}^{\text{FS}} \cup \tilde{\mathcal{E}}^{\text{SP}}$. However, obtaining $\tilde{\mathcal{E}}^{\text{JSD}}$ requires computing the similarity between all node pairs in $\tilde{\mathcal{V}}$ in every training epoch, which is time-consuming ($O(|\tilde{\mathcal{V}}|^2)$), and discovering the smallest values among them also requires additional computation. To alleviate such a complexity issue, we construct $k$-NN graphs [11, 37] of nodes in $\tilde{\mathcal{V}}$ based on the node feature similarity and structural proximity, and denote them as $\tilde{\mathcal{E}}_k^{\text{FS}}$ and $\tilde{\mathcal{E}}_k^{\text{SP}}$, respectively. Then, we compute the JSD values of the edges in $\tilde{\mathcal{E}}_k^{\text{FS}}$ and $\tilde{\mathcal{E}}_k^{\text{SP}}$ to obtain $\tilde{\mathcal{E}}_k^{\text{FS-JSD}}$ and $\tilde{\mathcal{E}}_k^{\text{SP-JSD}}$, instead of all edges in $\{(i,j)|i \in \tilde{\mathcal{V}}, j \in \tilde{\mathcal{V}}\}$ as in Eqn. 3, which notably alleviates the computation complexity from $O(|\tilde{\mathcal{V}}|^2)$ to $O(|\tilde{\mathcal{E}}_k^*|)$, where $|\tilde{\mathcal{V}}|^2 \gg |\tilde{\mathcal{E}}_k^*|$ for $* \in \{\text{SP, FS}\}$. That is, the graph augmentation is performed as: $\tilde{\mathcal{E}}^{\text{aug}} = \tilde{\mathcal{E}} \cup \tilde{\mathcal{E}}_k^{\text{FS-JSD}} \cup \tilde{\mathcal{E}}_k^{\text{SP-JSD}}$. For the implementation details, please refer to Appendix B.3.

Our proposed augmentation strategy is superior to existing works that utilize the graph properties [8, 12, 45] in terms of robustness and scalability. Detailed explanations and supporting results

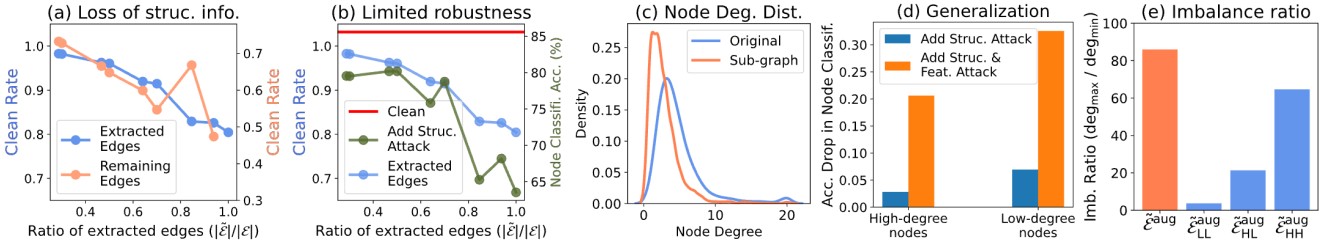

**Figure 2: (a)** Clean rate of the extracted edges and remaining edges over the ratio of extracted edges. **(b)** Node classification accuracy under structure attack and clean rate of extracted edges over the ratio of extracted edges. **(c)** Node degree distribution of original graph and extracted sub-graph. **(d)** Accuracy drop in node classification under attacks for high/low-degree nodes. **(e)** Imbalance ratio of $\tilde{\mathcal{E}}^{\text{aug}}$, $\tilde{\mathcal{E}}^{\text{aug}}_{\text{LL}}$, $\tilde{\mathcal{E}}^{\text{aug}}_{\text{HL}}$, and $\tilde{\mathcal{E}}^{\text{aug}}_{\text{HH}}$. Cora dataset is used. *Struc. Attack* indicates *metattack* 25% and *Feat. Attack* indicates Random Gaussian noise 50%.

can be found in Sec 5.3.3 and Appendix B.3. Furthermore, it is important to note that the sub-graph extraction process (Phase 1) including obtaining $\tilde{\mathcal{E}}^{*}_{k}$ is done offline before proceeding to Phase 2, which is described in Section 4.4, and thus Phase 1 does not increase the computation burden of Phase 2.

*4.4.2 Group-training Strategy (Phase 2-2).* To alleviate the imbalanced node degree distribution of the sub-graph, we balance the node degree distribution by splitting $\tilde{\mathcal{E}}^{\text{aug}}$ into three groups, i.e., $\tilde{\mathcal{E}}^{\text{aug}}_{\text{LL}}$, $\tilde{\mathcal{E}}^{\text{aug}}_{\text{HL}}$, and $\tilde{\mathcal{E}}^{\text{aug}}_{\text{HH}}$, and independently train the link predictor in the GSR module on each set. More precisely, $\tilde{\mathcal{E}}^{\text{aug}}_{\text{HH}}$ and $\tilde{\mathcal{E}}^{\text{aug}}_{\text{LL}}$ denote the set of edges incident to two high-degree nodes and two low-degree nodes, respectively, and $\tilde{\mathcal{E}}^{\text{aug}}_{\text{HL}}$ denotes the set of edges between a high-degree node and a low-degree node. Note that a node with its degree less than the median in the node degrees is considered as low-degree. To verify whether the splitting strategy balances the node degree distribution, we measure the imbalance ratio of the node degree distribution of edge set, which is defined as $I_{\text{ratio}} = \frac{\deg_{\max}}{\deg_{\min}}$, where $\deg_{\max}$ and $\deg_{\min}$ denote the maximum and minimum degrees in the node degree distribution, respectively [40]. Note that a large $I_{\text{ratio}}$ implies that the set is highly imbalanced. In Fig. 2(e), we observe that the imbalance ratios of $\tilde{\mathcal{E}}^{\text{aug}}_{\text{LL}}$, $\tilde{\mathcal{E}}^{\text{aug}}_{\text{HL}}$, and $\tilde{\mathcal{E}}^{\text{aug}}_{\text{HH}}$ are lower than that of $\tilde{\mathcal{E}}^{\text{aug}}$, which shows that the splitting strategy indeed balances the node degree distribution. We define the balanced link prediction loss by combining the link prediction loss in Eqn. 2 for each group, i.e., $L^{l}_{\tilde{\mathcal{E}}^{\text{aug}}_{\text{LL}}}$, $L^{l}_{\tilde{\mathcal{E}}^{\text{aug}}_{\text{HL}}}$, and $L^{l}_{\tilde{\mathcal{E}}^{\text{aug}}_{\text{HH}}}$, as follows: $L^{l}_{\mathcal{E}} = L^{l}_{\tilde{\mathcal{E}}^{\text{aug}}_{\text{LL}}} + L^{l}_{\tilde{\mathcal{E}}^{\text{aug}}_{\text{HL}}} + L^{l}_{\tilde{\mathcal{E}}^{\text{aug}}_{\text{HH}}}$. We argue that the link predictor in the GSR module is learned in a balanced manner in terms of the node degree distribution, which leads to the improvement of the generalization ability of the GSR module to low-degree nodes. Specifically, since the number of edges incident to each node is more evenly distributed in $\tilde{\mathcal{E}}^{\text{aug}}_{\text{LL}}$ and $\tilde{\mathcal{E}}^{\text{aug}}_{\text{HL}}$ than $\tilde{\mathcal{E}}^{\text{aug}}$, low-degree nodes are far more involved in computing $L_{\tilde{\mathcal{E}}^{\text{aug}}_{\text{LL}}}$ and $L_{\tilde{\mathcal{E}}^{\text{aug}}_{\text{HL}}}$ than $L_{\tilde{\mathcal{E}}^{\text{aug}}}$. Consequently, the combined loss, i.e., $L^{l}_{\mathcal{E}}$, is computed with more emphasis on low-degree nodes. Lastly, it is important to note that the message passing of the GSR module is performed on the whole edge sets, i.e., $\tilde{\mathcal{E}}^{\text{aug}}$, rather than the split sets. It is important to note that the above loss *does not increase the complexity of the model training nor the number of parameters*, because the number of samples used for training remains the same, and the parameters for the link predictor are shared among the groups. Moreover, we can

further split the edge set in a more fine-grained manner to obtain a more balanced edge sets, which will be later demonstrated in 5.3.4.

## 4.5 Training and Inference

**Training.** SG-GSR is trained to minimize the objective function: $L_{\text{final}} = L_{\tilde{\mathcal{V}}} + \lambda_{\mathcal{E}} \sum_{l=1}^{L} L^{l}_{\mathcal{E}}$, where $L_{\tilde{\mathcal{V}}}$ indicates the node classification loss on the set of labeled nodes in $\tilde{\mathcal{V}}$ as in Eqn. 1. Moreover, $L^{l}_{\mathcal{E}}$ and $\lambda_{\mathcal{E}}$ indicate the grouped link prediction loss for the $l$-th layer as in Eqn. 2 and the combination coefficient, respectively. During training, the parameters $\{\mathbf{W}^{l}\}_{l=1}^{L}$ of SG-GSR are updated to accurately predict labels of the nodes in $\tilde{\mathcal{V}}$ and clean links for each group, i.e., $\tilde{\mathcal{E}}^{\text{aug}}_{\text{HH}}$, $\tilde{\mathcal{E}}^{\text{aug}}_{\text{HL}}$, and $\tilde{\mathcal{E}}^{\text{aug}}_{\text{LL}}$.

**Inference.** In the inference phase, we use the knowledge obtained during training to refine the target attacked graph structure. More precisely, based on the learned model parameters $\{\mathbf{W}^{l}\}_{l=1}^{L}$, we compute the attention coefficients $\{\alpha^{l}_{ij}\}_{l=1}^{L}$ of the existing edges in the target attacked graph $\mathcal{E}$ followed by the message passing procedure as described in Section 4.1. In other words, we minimize the effect of adversarial edges during the message passing procedure, thereby achieving robustness against adversarial attacks on the graph structure.

## 5 EXPERIMENT

### 5.1 Experimental Settings

*5.1.1 Datasets.* We evaluate SG-GSR and baselines on **five existing datasets** (i.e., Cora [12], Citeseer [12], Pubmed [12], Polblogs [12], and Amazon [28]) and **two newly introduced datasets** (i.e., Garden and Pet) that are proposed in this work based on Amazon review data [10, 26] to mimic e-commerce fraud (Refer to Appendix D.2 for details). For each graph, we use a random 1:1:8 split for training, validation, and testing. The details of the datasets are given in Appendix D.1.

*5.1.2 Experimental Details.* We compare SG-GSR with a wide range of robust GNN baselines including GSR methods under poisoning structure and feature attacks, following existing robust GNN works [4, 12, 16, 17, 20, 22]. We consider three attack scenarios, i.e., structure attacks, structure-feature attacks, and e-commerce fraud. Note that in this work we mainly focus on graph modification attacks (i.e., modifying existing topology or node features). A discussion of robustness under graph injection attacks and adaptive

**Table 1: Node classification performance under non-targeted attack (i.e., *metattack*) and feature attack. OOM indicates out of memory on 12GB TITAN Xp. OOT indicates out of time (24h for each run is allowed).**

| Dataset | Setting | SuperGAT | RGCN | ProGNN | GEN | ELASTIC | AirGNN | SLAPS | RSGNN | CoGSL | STABLE | EvenNet | SE-GSL | SG-GSR |
|---|---|---|---|---|---|---|---|---|---|---|---|---|---|---|
| Cora | Clean | 85.4±0.3 | 84.0±0.1 | 82.9±0.3 | 83.9±0.8 | 85.5±0.4 | 83.6±0.3 | 75.1±0.2 | 85.1±0.3 | 80.2±0.3 | 85.1±0.3 | 85.5±0.4 | 85.5±0.3 | 85.5±0.1 |
| | + Meta 25% | 62.6±1.7 | 53.4±0.3 | 70.7±0.2 | 67.4±1.2 | 67.5±0.3 | 63.0±0.7 | 75.0±0.5 | 81.8±0.3 | 63.6±0.3 | 79.0±0.4 | 75.0±0.3 | 70.2±0.4 | **83.1±0.5** |
| | + Feat attack | 56.5±0.6 | 49.4±0.9 | 47.7±0.5 | 45.2±2.4 | 55.9±1.6 | 50.4±0.6 | 49.2±0.1 | 65.7±2.1 | 49.6±1.5 | 52.2±0.1 | 56.5±0.5 | 51.8±0.5 | **67.6±1.4** |
| Citeseer | Clean | 75.1±0.2 | 73.0±0.4 | 72.5±0.5 | 75.5±0.3 | 74.7±0.4 | 73.2±0.3 | 73.6±0.1 | 74.4±1.1 | **76.2±0.1** | 75.5±0.7 | 74.2±0.2 | 76.1±0.5 | 75.4±0.2 |
| | + Meta 25% | 64.5±0.6 | 58.6±0.9 | 68.4±0.6 | 71.9±0.8 | 66.3±1.0 | 62.2±0.6 | 73.1±0.6 | 73.9±0.7 | 71.6±0.7 | 73.4±0.3 | 71.6±0.3 | 70.3±0.8 | **75.2±0.1** |
| | + Feat attack | 57.1±0.9 | 50.3±0.6 | 52.6±0.2 | 50.4±1.1 | 60.8±1.3 | 58.1±1.1 | 52.3±0.4 | 64.0±0.3 | 57.4±1.5 | 58.4±0.4 | 59.7±0.4 | 59.0±0.9 | **66.8±1** |
| Pubmed | Clean | 84.0±0.5 | 86.9±0.1 | OOM | 86.5±0.5 | **88.1±0.1** | 87.0±0.1 | 83.4±0.3 | 84.8±0.4 | OOM | 85.5±0.2 | 87.5±0.2 | OOT | 87.6±0.2 |
| | + Meta 25% | 74.4±1.8 | 82.0±0.3 | OOM | 80.1±0.3 | 85.4±0.1 | 84.2±0.0 | 83.1±0.1 | 84.7±0.5 | OOM | 81.6±0.6 | 87.2±0.2 | OOT | **87.3±0.2** |
| | + Feat attack | 58.4±0.3 | 44.9±0.8 | OOM | 52.6±0.2 | 55.3±0.6 | 62.3±0.1 | 53.3±0.8 | 64.7±0.3 | OOM | 54.7±0.7 | 64.6±3.9 | OOT | **65.5±0.5** |
| Polblogs | Clean | 96.0±0.3 | 95.4±0.1 | 93.2±0.6 | 96.1±0.4 | 95.7±0.3 | 95.0±0.7 | 54.1±1.3 | 93.0±1.8 | 95.7±0.3 | 95.6±0.4 | 95.2±0.6 | 95.1±0.3 | **96.2±0.1** |
| | + Meta 25% | 79.6±2.0 | 66.9±2.2 | 63.2±4.4 | 79.3±7.7 | 63.6±1.5 | 57.3±4.4 | 52.2±0.1 | 65.0±1.9 | 51.9±0.2 | 75.2±3.4 | 59.1±6.1 | 68.3±1.2 | **87.8±0.7** |
| Amazon | Clean | 82.5±1.1 | 82.2±1.3 | OOM | 90.2±0.2 | 89.6±0.1 | 87.6±0.8 | 79.6±0.8 | 89.6±1.2 | OOM | 88.8±0.4 | 88.8±0.5 | OOT | **91.1±0.2** |
| | + Meta 25% | 76.0±1.6 | 73.2±0.7 | OOM | 85.6±0.9 | 86.7±0.2 | 85.6±0.4 | 79.0±0.3 | 86.9±1.6 | OOM | 81.7±0.3 | 85.4±1.5 | OOT | **89.2±0.2** |
| | + Feat attack | 75.2±0.5 | 71.1±2.3 | OOM | 85.1±0.6 | 85.4±0.3 | 83.3±0.2 | 71.8±0.6 | 85.0±1.5 | OOM | 79.5±0.8 | 85.3±0.8 | OOT | **87.2±0.4** |

**Table 2: Node classification performance under targeted attack (i.e., *nettack*) and feature attack.**

| Dataset | Setting | SuperGAT | RGCN | ProGNN | GEN | ELASTIC | AirGNN | SLAPS | RSGNN | CoGSL | STABLE | EvenNet | SE-GSL | SG-GSR |
|---|---|---|---|---|---|---|---|---|---|---|---|---|---|---|
| Cora | Clean | 83.1±1.0 | 81.5±1.1 | 85.5±0.0 | 82.7±3.5 | 86.4±2.1 | 79.9±1.1 | 70.7±2.3 | 84.3±1.0 | 76.3±0.6 | 85.5±1.0 | 85.1±1.6 | 85.5±1.0 | **86.4±1.1** |
| | + Net 5 | 60.6±2.8 | 55.8±0.6 | 67.5±0.0 | 61.5±3.9 | 67.5±2.1 | 61.0±2.5 | 68.7±2.6 | 73.1±1.5 | 61.5±0.6 | 76.3±1.5 | 66.3±1.2 | 67.5±0.6 | **77.1±1.7** |
| | + Feat attack | 59.4±2.5 | 52.6±0.6 | 57.8±0.0 | 47.0±2.0 | 63.1±1.8 | 54.2±1.0 | 39.0±3.0 | 71.9±0.6 | 46.6±0.6 | 64.7±1.5 | 60.6±2.0 | 59.0±0.5 | **72.7±1.1** |
| Citeseer | Clean | 82.5±0.0 | 81.0±0.0 | 82.5±0.0 | 82.5±0.0 | 82.5±0.0 | 82.5±1.3 | 81.5±0.8 | 84.1±0.0 | 81.5±0.8 | 82.5±0.0 | 82.5±0.0 | 82.5±0.0 | **85.7±2.2** |
| | + Net 5 | 54.5±4.9 | 50.3±3.7 | 71.5±0.0 | 77.3±1.5 | 79.9±0.9 | 70.4±2.0 | 81.0±1.3 | 78.8±2.0 | 79.9±0.8 | 82.5±0.0 | 79.9±2.1 | 82.5±0.0 | **83.1±0.8** |
| | + Feat attack | 49.2±1.3 | 47.1±3.3 | 68.3±0.0 | 40.2±4.6 | 57.7±4.0 | 52.9±3.0 | 70.9±2.7 | 74.6±2.6 | 46.0±2.6 | 65.1±3.4 | 63.5±4.8 | 77.8±1.5 | **83.1±2.7** |
| Pubmed | Clean | 87.6±0.4 | 89.8±0.0 | OOM | 89.8±0.0 | 90.5±0.3 | 90.9±0.2 | 80.5±1.5 | 88.4±0.5 | OOM | 89.3±0.5 | **90.9±0.5** | OOT | 90.9±1.9 |
| | + Net 5 | 70.6±0.7 | 70.1±0.3 | OOM | 72.0±0.0 | 85.8±0.3 | 83.0±0.9 | 80.5±1.5 | 87.8±1.3 | OOM | 83.3±0.3 | 72.2±0.7 | OOT | **88.0±0.3** |
| | + Feat attack | 70.4±1.2 | 61.5±1.3 | OOM | 61.8±0.0 | **78.1±0.8** | 77.1±0.9 | 56.6±2.0 | 76.5±0.7 | OOM | 73.1±0.7 | 67.9±0.9 | OOT | 75.8±2.0 |
| Polblogs | Clean | 97.7±0.4 | 97.4±0.2 | 97.1±0.3 | 97.8±0.2 | 97.8±0.3 | 97.3±0.2 | 54.1±1.3 | 96.4±0.7 | 96.9±0.2 | 97.5±0.2 | **97.9±0.4** | 97.0±0.4 | **97.9±0.2** |
| | + Net 5 | 95.9±0.2 | 93.6±0.5 | 96.1±0.6 | 94.8±1.2 | 96.2±0.3 | 90.0±0.9 | 51.4±3.4 | 93.4±0.7 | 89.6±0.6 | 96.1±0.4 | 94.2±1.1 | 95.1±0.3 | **96.5±0.2** |

attacks are included in Appendix C.5 and C.6. We describe the baselines, evaluation protocol, and implementation details in Appendix D.3, D.4, and D.5, respectively.

## 5.2 Evaluation of Adversarial Robustness

*5.2.1 Against non-targeted and targeted attacks.* We first evaluate the robustness of SG-GSR under *metattack*, a non-targeted attack. In Table 1, we have the following two observations: **1)** SG-GSR consistently outperforms all baselines under structure attack (i.e., + Meta 25%). We attribute the superiority of SG-GSR over multi-faceted methods to utilizing the clean sub-graph instead of the given attacked graph. Moreover, SG-GSR also surpasses all feature-based methods since the clean sub-graph and the graph augmentation strategy enrich the structural information, while also utilizing the given node features. **2)** SG-GSR consistently performs the best under structure-feature attacks (i.e., + Feat. Attack). We argue that leveraging the structural information (i.e., clean sub-graph) in addition to the node features alleviates the weakness of feature-based approaches. Moreover, we observe similar results against the targeted attack (i.e., *nettack*) in Table 2.

*5.2.2 Against e-commerce fraud.* We newly design two new benchmark graph datasets, i.e., Garden and Pet, where the node label is the product category, the node feature is bag-of-words representation of product reviews, and the edges indicate the co-review relationship between two products reviewed by the same user. While existing works primarily focus on artificially generated attack datasets, to the best of our knowledge, this is the first work proposing new datasets for evaluating the robustness of GNNs under adversarial attacks that closely imitate a real-world e-commerce system containing malicious fraudsters. Appendix D.2 provides a comprehensive description of the data generation process. In Table 3, we

**Table 3: Node classification performance under e-commerce fraud.**

| Methods | Garden | | Pet | |
|---|---|---|---|---|
| | Clean | + Fraud | Clean | + Fraud |
| SuperGAT | 86.0±0.4 | 81.8±0.3 | 87.3±0.1 | 80.6±0.3 |
| RGCN | 87.1±0.5 | 81.5±0.3 | 86.6±0.1 | 78.5±0.2 |
| ProGNN | OOT | OOT | OOT | OOT |
| GEN | 87.1±0.6 | 82.2±0.0 | 88.5±0.6 | 81.1±0.3 |
| ELASTIC | **88.4±0.1** | 82.9±0.1 | 88.9±0.1 | 81.3±0.2 |
| AirGNN | 87.1±0.2 | 80.9±0.4 | 88.5±0.1 | 79.3±0.3 |
| SLAPS | 79.3±0.8 | 74.6±0.2 | 81.4±0.2 | 75.8±0.2 |
| RSGNN | 81.8±0.6 | 76.3±0.0 | 81.6±0.4 | 74.2±0.0 |
| CoGSL | OOM | OOM | OOM | OOM |
| STABLE | 84.3±0.3 | 81.0±0.3 | 87.9±0.2 | 80.8±0.2 |
| EvenNet | 86.3±0.2 | 81.3±0.4 | 88.5±0.2 | 81.0±0.1 |
| SE-GSL | 82.0±0.4 | 77.3±0.6 | 87.9±0.4 | 77.5±0.7 |
| SG-GSR | 88.3±0.1 | **83.3±0.2** | **89.4±0.1** | **81.9±0.1** |

observe that SG-GSR outperforms the baselines under the malicious actions of fraudsters, which indicates that SG-GSR works well not only under artificially generated adversarial attacks, but also under attacks that are plausible in the real-world e-commerce systems.

## 5.3 Model Analyses

*5.3.1 Ablation studies on each component of SG-GSR.* To evaluate the importance of each component of SG-GSR, i.e., clean sub-graph extraction (*SE*), graph augmentation (*GA*), and the group-training strategy (*GT*), we add them one by one to a baseline model, i.e., SuperGAT. In Table 4, we have the following observations: **1)** Adding clean sub-graph extraction (*SE*) to SuperGAT is considerably helpful for defending against adversarial attacks, which indicates that the false positive issue when minimizing Eqn. 2 is alleviated by successfully extracting the clean sub-graph. **2)** Randomly adding edges

**Table 4: Ablation studies. *SE*, *GA*, and *GT* denote the sub-graph extraction, graph augmentation, and group-training, respectively. *Rand*, C, F, and S indicate whether *GA* considers random edge addition, class homophily, feature smoothness, and structural proximity, respectively.**

| Component | | | Cora | | | Citeseer | | |
|---|---|---|---|---|---|---|---|---|
| SE | GA | GT | Clean | + Meta 25% | + Feat. Attack | Clean | + Meta 25% | + Feat. Attack |
| ✗ | ✗ | ✗ | 84.3±0.5 | 62.6±1.7 | 56.5±0.6 | 74.2±0.2 | 64.5±0.6 | 57.1±0.9 |
| ✓ | ✗ | ✗ | 84.4±0.5 | 80.6±0.7 | 58.6±1.1 | 74.6±0.2 | 74.4±0.4 | 60.9±0.5 |
| ✓ | *Rand* | ✗ | 83.6±0.4 | 78.0±1.4 | 49.9±1.2 | 74.9±0.6 | 73.6±0.4 | 48.7±1.7 |
| ✓ | C | ✗ | 84.6±0.3 | 81.0±0.3 | 60.1±0.8 | 74.6±0.2 | 74.3±0.6 | 59.0±0.6 |
| ✓ | C, F | ✗ | 84.9±0.3 | 81.7±0.1 | 64.1±0.9 | 74.5±0.1 | 74.6±0.6 | 62.2±0.6 |
| ✓ | C, F, S | ✗ | 84.6±0.1 | 82.1±0.3 | 64.3±0.7 | 74.8±0.2 | 74.7±0.4 | 62.6±0.6 |
| ✓ | C, F, S | ✓ | **85.5±0.1** | **83.4±0.5** | **67.6±1.4** | **75.4±0.2** | **75.2±0.1** | **66.8±1.0** |

**Table 5: Ablation study on *SE*. Feat. Attack indicates Random Gaussian noise 50%.**

| | Pubmed | | | Amazon | | |
|---|---|---|---|---|---|---|
| Setting | Clean | + Meta 25% | + Feat. Attack | Clean | + Meta 25% | + Feat. Attack |
| *SE w/o* $\tilde{\mathcal{E}}^{st}$ | **87.6±0.2** | 87.2±0.1 | 64.4±0.3 | 90.6±0.1 | 88.5±0.4 | 86.6±0.6 |
| *SE w/o* $\tilde{\mathcal{E}}^{feat}$ | **87.6±0.2** | 83.8±0.2 | 66.0±0.4 | **91.1±0.2** | **89.2±0.2** | 86.8±0.6 |
| *SE* | **87.6±0.2** | **87.3±0.2** | **66.0±0.4** | **91.1±0.2** | **89.2±0.2** | **87.2±0.4** |

(*Rand*) to augment the extracted sub-graph performs worse than not performing any augmentation at all, and considering all three properties (i.e., class homophily (C), feature smoothness (F), and structural proximity (S)) yields the best performance. This implies that the randomly added edges contain edges that are not important for predicting the node labels that deteriorate the performance, while our proposed *GA* mainly adds important edges for predicting the node labels by considering various real-world graph properties. This demonstrates that the proposed graph augmentation strategy supplements the loss of structural information of the extracted sub-graph that is crucial for accurately predicting the node labels. Moreover, the augmented graph that satisfies various real-world graph properties enhances the generalization ability of the link predictor in the GSR module. **3)** Adding the group-training (*GT*) strategy significantly improves the node classification accuracy. We attribute this to the fact that *GT* allows the proposed link predictor to pay more attention to low-degree nodes during training, thereby enhancing the generalization ability to low-degree nodes.

*5.3.2 Further analysis on sub-graph extraction (SE).* To verify the benefit of jointly considering the structural proximity and the node feature similarity for extracting the clean sub-graph, we compare *SE* with *SE w/o* $\tilde{\mathcal{E}}^{feat}$, which only considers the structural proximity, and *SE w/o* $\tilde{\mathcal{E}}^{st}$, which only considers the node feature similarity. Note that *SE w/o* $\tilde{\mathcal{E}}^{st}$ is equivalent to the extraction method adopted by STABLE [17]. In Table 5, we observe that *SE* outperforms *SE w/o* $\tilde{\mathcal{E}}^{st}$ especially under the structure-feature attack. This is because when the node features are noisy or attacked, it becomes hard to distinguish clean edges from adversarial ones solely based on the node feature similarity, which aggravates the issue regarding false positives edges in the extracted sub-graph. The superior performance of *SE* implies that jointly considering both feature similarity and structural proximity is beneficial for alleviating false positive issue.

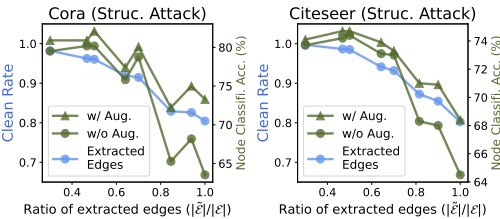

**Figure 3: Node classification accuracy with and without our proposed graph augmentation (*GA*) strategy. *Struc. Attack* indicates *metattack* 25%.**

**Table 6: Analysis on *GA* against noisy node label. *GA w. C* indicates that only class homophily is considered in *GA*. Note that *GA w. C* is equivalent to the augmentation strategy of GAUG [45] in which only the class homophily property is considered for graph augmentation.**

| | Cora | | | Citeseer | | |
|---|---|---|---|---|---|---|
| Noise rate | 0.1 | 0.2 | 0.3 | 0.1 | 0.2 | 0.3 |
| *GA w. C* | 80.9±0.1 | 78.9±0.9 | 76.0±1.1 | 73.8±0.7 | 72.9±0.2 | 69.3±1.3 |
| *GA w. C,F,S* | **82.1±0.7** | **79.9±0.5** | **76.5±0.4** | **74.6±0.6** | **74.3±0.5** | **71.9±0.6** |

*5.3.3 Further analysis on graph augmentation (GA).* As shown in Table 4, the proposed graph augmentation strategy effectively supplements the loss of structural information of the extracted sub-graph, which in turn improves the node classification performance. To be more concrete, in Fig. 3, we report the node classification accuracy with and without our proposed graph augmentation strategy over various ratios of extracted edges (i.e., $|\tilde{\mathcal{E}}|/|\mathcal{E}|$) under structural attacks (i.e., *metattack* 25%). We observe that the proposed augmentation strategy consistently improves the GSR module.

We further compare our proposed graph augmentation strategy (i.e., *GA w. C,F,S*) with *GA w. C* to verify the effectiveness of the proposed augmentation method when the label information contains noise. Note that *GA w. C* is equivalent to the augmentation strategy adopted by GAUG [45] in which only the class homophily property is considered for graph augmentation. We train each model on Cora and Citeseer with varying label noise rates, i.e., {0.1, 0.2, 0.3}, where the noise is injected by randomly assigning another label. In Table 6, we observe that *GA w. C,F,S* outperforms *GA w. C* under noisy node labels. This is because *GA w. C* solely relies on the uncertain node label predictions of the model when supplementing the structural information of the sub-graph, whereas *GA w. C,F,S* considers diverse properties in addition to the class homophily.

*5.3.4 Further analysis on the group-training (GT) strategy.* As mentioned in Sec. 4.4.2, we can further split the edge set in a more fine-grained manner to obtain a more balanced edge sets. Specifically, we compare the two splitting strategies, L-H and L-M-H. L-H indicates that we split the edge set into three groups, $\tilde{\mathcal{E}}_{LL}^{aug}$, $\tilde{\mathcal{E}}_{HH}^{aug}$, and $\tilde{\mathcal{E}}_{HL}^{aug}$, where L and H indicate low- and high-degree nodes. L-M-H indicates that we split the edge set into six groups $\tilde{\mathcal{E}}_{LL}^{aug}$, $\tilde{\mathcal{E}}_{MM}^{aug}$, $\tilde{\mathcal{E}}_{HH}^{aug}$, $\tilde{\mathcal{E}}_{ML}^{aug}$, $\tilde{\mathcal{E}}_{HL}^{aug}$, and $\tilde{\mathcal{E}}_{HM}^{aug}$, where L, M, and H indicate low-, mid-, and high-degree nodes. In Table 7, we observe that adding the group-training (*GT*) strategy significantly improves the node classification accuracy. Furthermore, splitting the edge set in a more

**Table 7: Further ablation studies on *GT*. *SE, GA*, and *GT* denote the sub-graph extraction module, graph augmentation, and group-training, respectively. Moreover, L, M, and H indicate low-, mid-, and high-degree nodes. *Feat. Attack* indicates Random Gaussian noise 50%.**

| Component | | | Cora | | | Citeseer | | |
|---|---|---|---|---|---|---|---|---|
| SE | GA | GT | Clean | + Meta 25% | + Feat. Attack | Clean | + Meta 25% | + Feat. Attack |
| ✓ | ✓ | ✗ | 84.6±0.1 | 82.1±0.3 | 64.3±0.7 | 74.8±0.2 | 74.7±0.4 | 62.6±0.6 |
| ✓ | ✓ | L-H | 84.7±0.6 | 82.8±0.4 | 65.8±1.0 | 75.1±0.3 | 74.9±0.2 | 65.2±0.3 |
| ✓ | ✓ | L-M-H | **85.5±0.1** | **83.4±0.5** | **67.6±1.4** | **75.4±0.2** | **75.4±0.3** | **66.8±1.0** |

**Table 8: Ablation studies on the group-training (*GT*) strategy on high/low-degree nodes. *Feat. Attack* indicates Random Gaussian noise 50%.**

| Dataset | Attack | Node | SG-GSR w/o *GT* | SG-GSR | Diff.(%) |
|---|---|---|---|---|---|
| Cora | Clean | high-degree | 85.6±0.5 | **86.7±0.2** | 1.1 |
| | | low-degree | 83.8±0.1 | **84.5±0.6** | 0.7 |
| | + Meta 25% | high-degree | 84.4±0.8 | **86.7±0.8** | 2.3 |
| | | low-degree | 78.2±0.8 | **80.7±0.1** | 2.5 |
| | + Feat. Attack | high-degree | 73.7±1.3 | **76.0±1.5** | 2.3 |
| | | low-degree | 56.9±1.1 | **61.3±1.4** | 4.4 |
| Citeseer | Clean | high-degree | 76.4±0.2 | **77.7±0.4** | 1.3 |
| | | low-degree | 73.0±0.4 | **73.2±0.4** | 0.2 |
| | + Meta 25% | high-degree | 77.9±0.7 | **78.6±0.2** | 0.7 |
| | | low-degree | 70.8±0.3 | **72.1±0.4** | 1.3 |
| | + Feat. Attack | high-degree | 71.9±0.6 | **73.9±1.1** | 2.0 |
| | | low-degree | 54.0±0.2 | **60.3±1.0** | 6.3 |

fine-grained way, i.e., L-M-H, performs the best. We attribute this to the fact that more fine-grained *GT* allows the the node degree distribution to be more balanced, hence enhancing the generalization ability to low-degree nodes.

To further investigate the effectiveness of our proposed group-training strategy, we conduct an ablation study of *GT* with respect to the node degrees. In Table 8, we indeed observe that adding *GT* is more beneficial to low-degree nodes than to high-degree nodes in terms of robustness under attacks. We again attribute this to the fact that GT allows the proposed link predictor to pay more attention to low-degree nodes during training, hence enhancing the generalization ability to low-degree nodes.

Moreover, we compare the performance of SG-GSR with the baselines (i.e., Tail-GNN, SLAPS, and RSGNN) that improve the performance on low-degree nodes. In Fig. 4, we have the following observations: **1)** SG-GSR outperforms Tail-GNN with a large gap under attacks since Tail-GNN is designed assuming when the given graph is clean, which suffers from a significant performance drop under attacks. **2)** SG-GSR consistently outperforms the feature-based methods, i.e., SLAPS and RSGNN, on low-degree nodes under both structure and structure-feature attacks. We attribute the effectiveness of SG-GSR on low-degree nodes to the group-training strategy that enhances the robustness of GSR to the low-degree nodes. Moreover, enriching the structural information as in SG-GSR is beneficial to mitigating the weakness of feature-based approaches under the structure-feature attacks. **3)** SG-GSR also outperforms the feature-based methods on high-degree nodes under both structure and structure-feature attacks. We conjecture that since the feature-based methods cannot fully exploit the abundant structural information that exist in high-degree nodes, i.e., neighboring nodes, their performance on high-degree nodes is limited.

*5.3.5 Sensitivity analysis.* We analyze the sensitivity of $\lambda_{\mathcal{E}}$, $\lambda_{\text{aug}}$, $k$, degree split, and hyperparameters in node2vec. For the results, please refer to Appendix C.2.

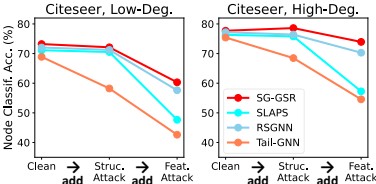

**Figure 4: Node Classification on low-/high-degree nodes on Citeseer dataset. *Struc. Attack* indicates *metattack* 25% and *Feat. Attack* indicates Random Gaussian noise 50%.**

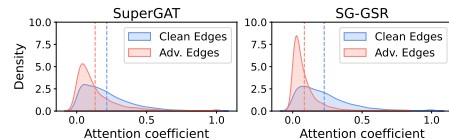

**Figure 5: Distribution of attention coefficients of clean and adversarial edges on the refined graph under *metattack*. Dashed lines indicate average values. Cora dataset is used.**

*5.3.6 Complexity analysis.* The complexity analysis of SG-GSR can be found in Appendix C.3.

## 5.4 Analysis of Refined Graph

In this subsection, we qualitatively analyze how well SG-GSR refines the attacked graph structure. Based on the learned model parameters $\{\mathbf{W}^l\}_{l=1}^{L}$, we compute the attention coefficients of the existing edges in the target attacked graph $\mathcal{E}$ in the last layer, i.e., $\alpha_{ij}^L$, to obtain the refined graph structure. In Fig. 5, we compare the distribution of attention coefficient values between the adversarial edges and the original clean edges. We clearly observe that the attention coefficients of adversarial edges are mostly assigned to values close to zero, whereas those of clean edges tend to be assigned to larger values. The result is further emphasized when comparing it to our backbone network, SuperGAT. This indicates that SG-GSR successfully minimizes the effect of adversarial edges during the message passing procedure, which enhances the robustness of GSR under the structural attacks. Further analyses of refined graphs are provided in Appendix C.4.

## 6 CONCLUSION

In this paper, we have discovered that existing GSR methods are limited by narrow assumptions, such as assuming clean node features, moderate structural attacks, and the availability of external clean graphs, resulting in the restricted applicability in real-world scenarios. To mitigate the limitations, we propose SG-GSR , which refines the attacked graph structure through the self-guided supervision regarding clean/adversarial edges. Furthermore, we propose a novel graph augmentation and group-training strategies in order to address the two technical challenges of the clearn sub-graph extraction, i.e., loss of structural information and imbalanced node degree distribution. We verify the effectiveness of SG-GSR through extensive experiments under various artificially attacked graph datasets. Moreover, we introduce novel graph benchmark datasets that simulate real-world fraudsters' attacks on e-commerce systems, which fosters a practical research in adversarial attacks on GNNs.

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

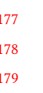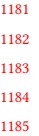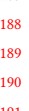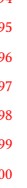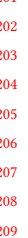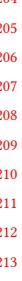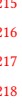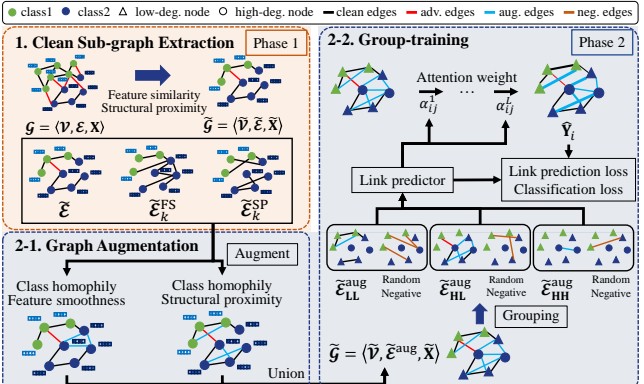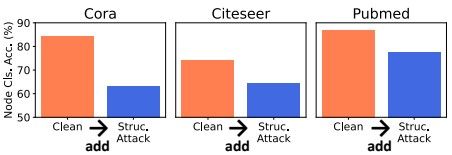

Figure 6: Overall architecture of SG-GSR.

## A OVERALL ARCHITECTURE

Fig. 6 shows the overall architecture of SG-GSR, and the detailed algorithm is provided in Algorithm. 1.

Figure 7: Performance of SuperGAT trained on a clean graph or an attacked graph on Cora, Citeseer, and Pubmed datasets. *Struc. Attack* indicates *metattack* 25%.

## B FURTHER DISCUSSION ON SG-GSR

### B.1 Discussion on Graph Structure Refinement Module (Sec. 4.1)

As mentioned in Sec. 4.1 of the main paper, by minimizing $L_{\mathcal{E}}^l$ (Eqn. 2), we expect that a large $e_{ij}^l$ is assigned to clean edges, whereas a small $e_{ij}^l$ is assigned to adversarial edges, thereby minimizing the effect of adversarial edges. However, since $\mathcal{E}$ may contain unknown adversarial edges due to structural attacks, the positive samples in Eqn. 2 may contain false positives, which leads to a sub-optimal solution under structural attacks. In Fig. 7, we indeed observe that the performance of SuperGAT drops significantly when it is trained on a graph after the structure attack (i.e., *metattack* 25%), which highlights the necessity of introducing the clean sub-graph extraction module.

### B.2 Discussion on Extraction of Clean Sub-graph (Sec. 4.2)

**Regarding the Importance of Considering Structural Proximity** To illustrate the effectiveness of our proposed clean sub-graph extraction method, we measure and visualize the cleanness of the extracted sub-graph in Fig. 8 over various $\lambda_{sp}$ and $\lambda_{fs}$ values. When only the graph structure is attacked (Fig. 8(a)), we observe that the extracted sub-graph gets cleaner as we extract more confidently

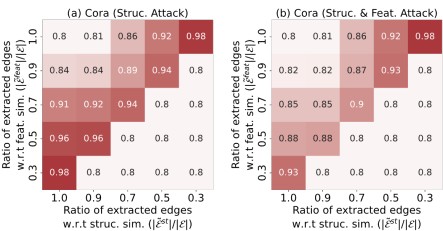

Figure 8: The cleanness of the extracted sub-graph obtained by the proposed sub-graph extraction method on Cora. Each element indicates the ratio of clean edges among the extracted edges. Dark color indicates that the ratio is high. *Struc. Attack* indicates *metattack* 25% and *Feat. Attack* indicates Random Gaussian noise 50%.

Table 9: Node classification accuracy (%) under the out-of-distribution (OOD) setting. $A \rightarrow B$ denotes training a GNN model on the A graph and evaluating its performance on the B graph. OG, Atk, and Sub represent the original clean graph, original attacked graph (*metattack* 25%), and extracted sub-graph, respectively. A high node classification accuracy indicates minimal distribution shift between the two graphs.

| | Cora | | | Citeseer | | |
|---|---|---|---|---|---|---|
| | OG → OG | Sub → OG | Atk → OG | OG → OG | Sub → OG | Atk → OG |
| GCN | 83.8 | 81.5 | 43.2 | 71.9 | 71.3 | 45.3 |
| GAT | 83.9 | 80.8 | 54.7 | 73.8 | 73.7 | 63.5 |

clean edges in terms of the node feature similarity. For example, as the ratio of extracted edges w.r.t feature similarity (i.e., $y$-axis) decreases from 1.0 to 0.3, the cleanness of the extracted sub-graph notably increases to 0.98. This implies that even leveraging only the node feature similarity well distinguishes the clean edges from the attacked structure. However, when the node features are also noisy or attacked (Fig. 8(b)), we observe that the distinguishability based on the feature similarity drops significantly. For example, when the ratio of extracted edges w.r.t feature similarity is 0.5, the cleanness of the extracted sub-graph drops from 0.96 to 0.88. This implies that when the node features are also noisy or attacked, extracting a sub-graph based only on the node feature similarity aggravates the issue regarding false positives edges. Hence, we argue that jointly considering both feature similarity and structural proximity is beneficial for alleviating the issue, because the structural proximity is helpful for distinguishing the clean edges even under the structure attack. For example, in Fig. 8(a) and (b), we observe that as we extract more confident edges based on structural proximity (i.e., from left to right on $x$-axis), the ratio of clean edges among the extracted edges increases. Our argument is corroborated by model analysis shown in Table 5 in Sec. 5.3.2. Note that a recent relevant work, called STABLE [17], only considers the node feature similarity, thereby being deteriorated when the node features are noisy or attacked.

**Regarding the Challenge of Clean sub-graph** In Sec. 4.3, we discover the two technical challenges of extracting the clean sub-graph that limit the robustness of the proposed GSR method: 1) loss of

structural information (**Section 4.3.1**), and 2) imbalanced node degree distribution (**Section 4.3.2**). However, it is worth considering that the extracted sub-graph may also exhibit out-of-distribution (OOD) characteristics compared to the original clean graph, as a significant number of edges are removed, altering the graph's context.

To investigate the potential of the extracted sub-graph being an OOD graph, we follow the evaluation protocol of a recent work on graph OOD [39]. First, given an input graph, we extract a clean sub-graph, and train a GNN model on the extracted sub-graph. Then, we use the trained GNN to perform inference on the nodes of the input graph (i.e., $Sub \rightarrow OG$). Second, we train another GNN model on the input graph, and perform inference on the nodes of the input graph (i.e., $OG \rightarrow OG$). Our assumption is that a significant performance gap between the two GNN models implies that the extracted sub-graph deviates from the original graph, which means it is an OOD graph. In Table 9, we observe that the performance gap between $OG \rightarrow OG$ and $Sub \rightarrow OG$ is negligible compared to the gap between $OG \rightarrow OG$ and $Atk \rightarrow OG$, indicating that the distribution of the extracted sub-graph closely resembles that of the original graph. Note that $Atk \rightarrow OG$ denotes training on an attacked input graph and performing inference on the nodes of the non-attacked input graph. This observation underscores that our proposed sub-graph extraction module preserves the content of the original graph, while effectively detecting and removing adversarial edges.

In addition to the challenges of clean sub-graph extraction on Cora dataset (Fig. 2(a), (b), (c), and (d) in the main paper), we provide results on Citeseer, Pubmed, and Polblogs datasets in Fig. 16, 17, and 18, respectively, all of which show similar results.

**Regarding the Existence of Labeled Nodes in Sub-graph** It is worth considering whether the extracted sub-graph contains the training nodes, especially in cases where the sub-graph is small or there is a scarcity of training nodes in the original graph. Please note that $\lambda_{fs}$ and $\lambda_{sp}$ determine the size of the extracted sub-graph. More precisely, setting a smaller $\lambda_{fs}$ and $\lambda_{sp}$ extracts smaller a sub-graph. Table 10 represents the number of nodes and edges in the extracted sub-graph and their number of labeled nodes when SG-GSR is trained with the best hyperparameters. From the results, we argue that the extracted sub-graph are large enough to contain the training nodes.

Furthermore, to confirm the case that the training nodes in the graph are scarce, we vary the label rate from 10% to 1% and train SG-GSR. Fig 9 represents the number of labeled nodes in the extracted sub-graph. Each point represents a sub-graph extracted using specific $\lambda_{fs}$ and $\lambda_{sp}$ values, where we vary $\lambda_{fs}$ and $\lambda_{sp}$ from 1 to 0.01. The points with star marker indicate the sub-graph extracted using the hyperparameters, i.e., $\lambda_{fs}$ and $\lambda_{sp}$, that yield the best performance. We observe that unless the extracted subgraph is extremely small (e.g., when only 2% of the entire nodes remain for 10% label rate), the extracted sub-graph always contains labeled nodes. Even under very scarce label rate, similar results are observed (e.g., when only 10% of the entire nodes remain for 1% label rate). It is important to further emphasize that, in our implementation, we imposed constraints on the size of the sub-graph, ensuring that the number of edges in the sub-graph adhered to a ratio of 30% in relation to the entire edge sets. By doing so, we made sure that each sub-graph always contains labeled nodes.

**Table 10: Statistics of the extracted sub-graph by SG-GSR given the attacked graph, where *metattack* 25% is used as the attack.**

| Dataset | Graph | # edges | # nodes | # labeled nodes |
|---------|-------|---------|---------|-----------------|
| Cora | Original graph | 5,069 | 2,485 | 249 |
| | Attacked graph | 6,336 | 2,485 | 249 |
| | Extracted Sub-graph | 3,125 | 2,178 | 221 |
| Citeseer | Original graph | 2,110 | 3,668 | 211 |
| | Attacked graph | 2,110 | 4,585 | 211 |
| | Extracted Sub-graph | 2,275 | 1,777 | 180 |

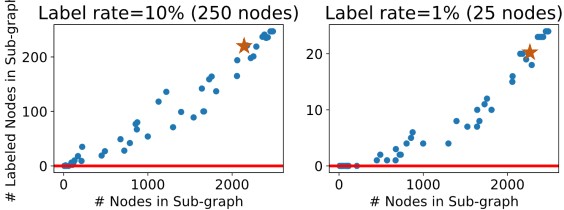

**Figure 9: The number of labeled nodes and the number of nodes in the extracted sub-graph according to the label rates. Star denotes the point that SG-GSR achieves the best performance.**

**Table 11: Execution time of node2vec.**

| Dataset | # Nodes | Time (sec) |
|---------|---------|------------|
| Cora | 2,485 | 23 |
| Citeseer | 2,110 | 18 |
| Pubmed | 19,717 | 219 |
| Polblogs | 1,222 | 11 |
| Amazon | 13,752 | 152 |

**Regarding the Time Complexity of node2vec** The execution time of node2vec is shown in the Table 11. Note that we use an efficient node2vec package (i.e., fastnode2vec [1]). As stated in the node2vec [9] paper and our implementation, it can be observed that the time complexity scales linearly with respect to the number of nodes. Since this process only needs to be performed once before the model training, it is considered an acceptable level of complexity regarding the importance of the structural proximity in extracting the clean sub-graph.

## B.3 Discussion on Graph Augmentation (Sec. 4.4.1)

**Implementation Details** As aforementioned in Sec. 4.4.1, we perform augmentations on the extracted sub-graph (i.e., $\tilde{\mathcal{E}}$) as follows:

$$\tilde{\mathcal{E}}^{aug} = \tilde{\mathcal{E}} \cup \tilde{\mathcal{E}}^{JSD} \cup \tilde{\mathcal{E}}^{FS} \cup \tilde{\mathcal{E}}^{SP}. \tag{4}$$

However, obtaining $\tilde{\mathcal{E}}^{JSD}$ requires computing the similarity between all node pairs in $\tilde{\mathcal{V}}$ in every training epoch, which is time-consuming ($O(|\tilde{\mathcal{V}}|^2)$), and discovering the smallest values among them also requires additional computation. To alleviate such complexity issue, we construct $k$-NN graphs [11, 37] of nodes in $\tilde{\mathcal{V}}$ based on the node feature similarity and structural proximity, and denote them as $\tilde{\mathcal{E}}_k^{FS}$ and $\tilde{\mathcal{E}}_k^{SP}$, respectively. Then, we compute the

JSD values of the edges in $\tilde{\mathcal{E}}_k^{\text{FS}}$ and $\tilde{\mathcal{E}}_k^{\text{SP}}$ instead of all edges in $\{(i, j) | i \in \tilde{\mathcal{V}}, j \in \tilde{\mathcal{V}}\}$ as in Eqn. 3, and add the edges with the smallest JSD values, denoted as $\tilde{\mathcal{E}}_k^{\text{FS-JSD}}$ and $\tilde{\mathcal{E}}_k^{\text{SP-JSD}}$, to the extracted sub-graph $\tilde{\mathcal{E}}$, which satisfy both feature smoothness/structural proximity and class homophily. Note that the number of added edges is set to $\lfloor |\tilde{\mathcal{E}}| \cdot \lambda_{\text{aug}} \rfloor$ for both $\tilde{\mathcal{E}}_k^{\text{FS-JSD}}$ and $\tilde{\mathcal{E}}_k^{\text{SP-JSD}}$, where $\lambda_{\text{aug}} \in [0, 1]$ is a hyperparameter. Hence, $\tilde{\mathcal{E}}^{\text{aug}}$ defined in Eqn. 4 is reformulated as:

$$\tilde{\mathcal{E}}^{\text{aug}} = \tilde{\mathcal{E}} \cup \tilde{\mathcal{E}}_k^{\text{FS-JSD}} \cup \tilde{\mathcal{E}}_k^{\text{SP-JSD}} \tag{5}$$

**Comparison with the Existing Graph Augmentation Methods**
We further analyze the effectiveness of our proposed augmentation strategy compared with existing approaches [8, 12, 45]. GAUG [45] adds class homophilic edges to better predict node labels. However, we argue that adding only class homophilic edges introduces bias and uncertainty to the model when the model predictions can be easily misestimated (e.g., noisy node label). This is because the structural information to be supplemented solely relies on the uncertain predictions of node labels. On the other hand, our proposed augmentation strategy considers diverse properties (i.e., feature smoothness and structural proximity) in addition to the class homophily thereby alleviating the issue incurred by relying solely on the class homophily, which is demonstrated in Sec. 5.3.3 and Table 6. ProGNN [12] and SLAPS [8] utilize the feature smoothness property, but they indeed require a model training process with heavy computation and memory burden to obtain an augmented graph. In contrast, our proposed strategy is more scalable than these methods [8, 12], since it does not require any model training process, and besides, the $k$-NN graph in terms of node features and structural features, i.e., $\tilde{\mathcal{E}}_k^{\text{FS}}$ and $\tilde{\mathcal{E}}_k^{\text{SP}}$, can be readily computed before the model training (i.e., Phase 1), which removes any additional computation burden during training.

## C ADDITIONAL EXPERIMENTAL RESULTS.

### C.1 Limitations of the Recent GSR

In Sec. 4.3.1, we demonstrated that our proposed sub-graph extraction module faces a challenge of losing vital structural information. In fact, as mentioned in Sec. 2.1, STABLE also employs a similar approach to our proposed sub-graph extraction module that detects and removes adversarial edges to extract clean edges from the attacked graph. In this section, we show that the clean sub-graph extraction module of STABLE also encounters the same problem that our proposed module faces, to corroborate the necessity of our novel graph augmentation strategy that reflects the real-world graph properties, i.e., class homophily, feature smoothness, and structural proximity.

In Fig. 10(a) and 11(a), we observe that as the ratio of extracted edges gets smaller, i.e., as the ratio of removed edges gets larger, most of the extracted edges are clean (blue line), but at the same time the remaining edges include a lot of clean edges as well (orange line), which incurs the loss of vital structural information. As a result, in Fig. 10(b) and 11(b), we observe that the robustness of STABLE is considerably restricted due to the lack of vital structural information. Note that we showed the same figure as Fig. 10 and 11 in terms of our proposed sub-graph extraction module in Fig. 2 of the main paper. Although both STABLE and our proposed sub-graph extraction module face the same challenge, which restricts

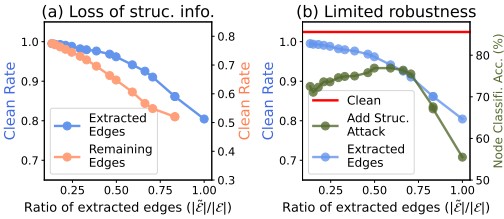

**Figure 10: The loss of structural information of STABLE on Cora.** *Struc. Attack* indicates *metattack 25%*.

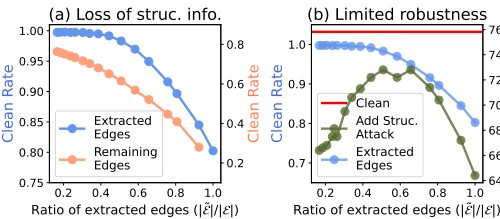

**Figure 11: The loss of structural information of STABLE on Citeseer.** *Struc. Attack* indicates *metattack 25%*.

their robustness, STABLE does not address this issue, while our proposed method successfully overcomes it through a novel graph augmentation strategy.

### C.2 Sensitivity Analysis

We analyze the sensitivity of six hyperparameters $\lambda_{\mathcal{E}}$, $\lambda_{\text{aug}}$, $k$, degree split, and hyperparameters in node2vec, i.e., $p$ and $q$.

- For $\lambda_{\mathcal{E}}$, we increase $\lambda_{\mathcal{E}}$ value from {0.2, 0.5, 1, 2, 3, 4, 5}, and evaluate the node classification accuracy of SG-GSR under structure attacks (*metattack 25%*). In Fig. 12, we observe that the accuracy of SG-GSR tends to increase as $\lambda_{\mathcal{E}}$ increases on both Cora and Citeseer datasets. In other words, assigning a higher weight to the link prediction loss during model training tends to yield a better performance. We argue that our proposed clean sub-graph extraction and graph augmentation method provide the clean and informative sub-graph to the link predictor as input edges, which makes the link predictor play an important role in GSR.

- In Fig 13(a) and (b), for any $\lambda_{\text{aug}}$ and $k$ in the graph augmentation module, SG-GSR consistently outperforms the sota baseline, RSGNN. This implies that our proposed graph augmentation module supplements the loss of structural information while not being sensitive to the hyperparameters.

- For degree split strategy, a node with its degree less than the median (i.e., 5:5 ratio) in the node degrees is considered as low-degree in our implementation. We further consider the splitting rule of 2:8, 3:7, 4:6, 5:5, 6:4, 7:3, and 8:2. Note that the low-degree node sets determined by 2:8 has the smallest node set. In Fig 13(c), we observe that SG-GSR consistently outperforms RSGNN, except for 2:8. We observe that in 2:8, the number of edges in the LL group is significantly small compared to the HH and HL groups, since a small number of nodes are assigned

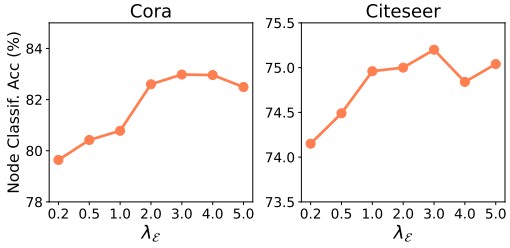

**Figure 12: Sensitivity Analysis on $\lambda_{\mathcal{E}}$.** We conduct the experiments under *metattack* 25%.

to low-degree nodes. It indicates that only a small set of low-degree nodes can take advantage of the group training strategy, which results in the performance degradation.

- For the hyperparameters in node2vec, we tune the in-out and return hyperparameters over $p, q \in \{0.5, 1, 2\}$. In Fig 13(d), we observe that SG-GSR consistently outperforms RSGNN, which indicates that the structural features obtained from node2vec are helpful for extracting clean sub-graphs while not being dependent on the hyperparameters of node2vec.
- We explore the sensitivity of $\lambda_{sp}$ and $\lambda_{fs}$ for SG-GSR. The performance variation of SG-GSR is displayed in Fig 14. It is worth noting that SG-GSR w/o sub-graph extraction (i.e., $\lambda_{sp} = 1$ and $\lambda_{fs} = 1$) consistently underperforms the SG-GSR w/ sub-graph extraction (i.e., $\lambda_{sp} < 1$ or $\lambda_{fs} < 1$)), which underscores the importance of the sub-graph extraction module.

### C.3 Complexity Analysis

In this subsection, we present a complexity analysis on training SG-GSR (Phase 2). Specifically, SG-GSR requires $O(C \cdot (|\tilde{\mathcal{E}}_k^{FS}| + |\tilde{\mathcal{E}}_k^{SP}|))$ for computing the JSD values in graph augmentation ($GA$), which is described in Sec. 4.4.1. Note that obtaining $\tilde{\mathcal{E}}_k^*$ is done offline before proceeding to Phase 2, and thus it does not increase the computation burden of Phase 2. For group-training strategy ($GT$), which is described in Sec. 4.4.2, SG-GSR requires $O(F^l \cdot |\tilde{\mathcal{E}}^{aug} \cup \tilde{\mathcal{E}}^{-, aug}|)$ for computing the grouped link prediction loss (i.e., $L_{\mathcal{E}}^l = L_{\tilde{\mathcal{E}}_{LL}^{aug}}^l + L_{\tilde{\mathcal{E}}_{HL}^{aug}}^l + L_{\tilde{\mathcal{E}}_{HH}^{aug}}^l$), where $\tilde{\mathcal{E}}^{aug}$ and $\tilde{\mathcal{E}}^{-, aug}$ denote positive and negative samples in link predictor, respectively, and $F^l$ is the dimensionality at the layer $l$. Note that $GT$ has equivalent time and space complexity as without $GT$, because the number of samples used for training remains the same, that is $|\tilde{\mathcal{E}}_{HH}^{aug} \cup \tilde{\mathcal{E}}_{HL}^{aug} \cup \tilde{\mathcal{E}}_{LL}^{aug}| = |\tilde{\mathcal{E}}^{aug}|$. Moreover, the GSR module of SG-GSR, which is described in Sec. 4.1, has equivalent time and space complexity as GAT [32].

Furthermore, we compare the training time of SG-GSR with the baselines to verify the scalability of SG-GSR. In Table 12, we report the training time per epoch on Cora dataset for all models. We observe that SG-GSR requires much less training time than ProGNN, GEN, RSGNN, and CoGSL, but requires more training time than SuperGAT and SLAPS. For GEN and CoGSL, which are multi-faceted methods, we argue that SG-GSR utilizes multi-faceted information far more efficiently than GEN and CoGSL. Regarding SuperGAT, which is our backbone network, SG-GSR significantly improves the performance of SuperGAT with acceptable additional

**Table 12: Training time comparison per epoch on Cora dataset (sec/epoch).**

|  | SuperGAT | ProGNN | GEN | SLAPS | RSGNN | CoGSL | SG-GSR |
|---|---|---|---|---|---|---|---|
| sec/epoch | 0.035 | 3.565 | 8.748 | **0.023** | 0.114 | 1.024 | 0.057 |

**Table 13: Node classification performance under graph injection attack (i.e., AFGSM [34]).**

|  | Cora | | Citeseer | | Pubmed | |
|---|---|---|---|---|---|---|
| Setting | Clean | + GIA | Clean | + GIA | Clean | + GIA |
| EvenNet | 90.0±0.0 | 46.7±3.3 | 68.3±3.3 | **51.7±3.3** | 91.3±0.7 | 53.3±3.3 |
| SG-GSR | **93.3±2.4** | **60.0±0.0** | **70.0±0.0** | 43.3±4.7 | **92.3±0.5** | **82.3±1.9** |

training time. Although SLAPS requires less training time, SG-GSR consistently outperforms SLAPS by utilizing multi-faceted information with acceptable additional training time. In summary, SG-GSR outperforms the baselines with acceptable training time.

### C.4 Further analysis on Refined Graph

In this subsection, we further analyze how well SG-GSR refines the attacked graph structure. We investigate whether SG-GSR can recover the communities, because successful adversarial attacks are known to add edges to destroy the community structures [12]. Specifically, we compute the inter-class (i.e., $\frac{\text{# inter-class edges}}{\text{# all existing edges}}$) and inter-community (i.e., $\frac{\text{# inter-community edges}}{\text{# all existing edges}}$) edge ratio where the communities are predefined by Spectral Clustering under the clean structure. In Fig. 15, we observe that although the structure attacks significantly increase the inter-class/community edge ratios, SG-GSR effectively recovers the community structure by the GSR module. This again corroborates that SG-GSR minimizes the effect of malicious inter-class/community edges that deteriorate the predictive power of GNNs, thereby enhancing the robustness against structure attack.

### C.5 Robustness against Graph Injection Attack

In this work, we mainly focused on graph modification attacks, which modify existing graph structures or node features. On the other hand, recent studies have shown that another type of attack, i.e., graph injection attacks (GIAs), can significantly reduce the performance of GNNs, even when only a few nodes are injected into the existing graph with limited resources [34, 49].

To further verify the robustness of SG-GSR against GIAs, we adopt the poisoning GIA method, i.e, AFGSM [34], as the attack method, and evaluate SG-GSR on the attacked graph. 20 nodes are randomly selected as our target nodes to be attacked. We use the default parameter settings in the authors' original implementation [34]. We compare the defense performance of SG-GSR with EvenNet, which is the current state-of-the-art robust GNN model, on Cora, Citeseer, and Pubmed datasets. Although GIAs are not the main focus of SG-GSR, we observe from Table 13 that SG-GSR remains competitive against EvenNet.

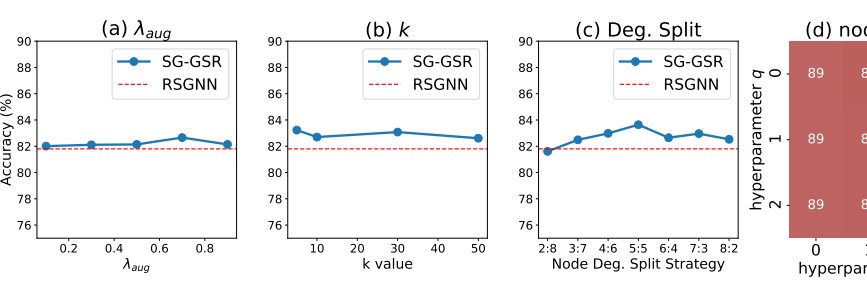
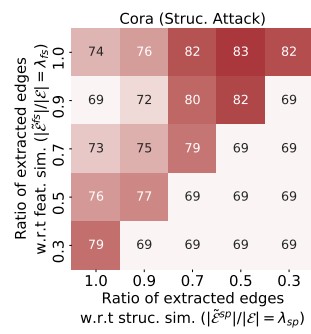

Figure 13: Sensitivity analysis on several hyperparameters. (a) Node classification accuracy over $\lambda_{\mathbf{aug}}$ on Cora dataset. (b) Node classification accuracy over $k$ on Cora dataset. (c) Node classification accuracy over various degree split strategies on Cora dataset. (d) Node classification accuracy over hyperparameters $p$ and $q$ in node2vec on Amazon dataset. Red-white-blue means outperformance, on-par, and underperformance compared with RSGNN, respectively.

Figure 14: Node classification accuracy over $\lambda_{\mathbf{fs}}$ and $\lambda_{\mathbf{sp}}$ on Cora dataset.

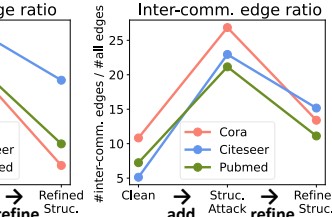

Figure 15: Comparison of inter-class/community edge ratios under a structural attack (i.e., *metattack* 25%) on Cora, Citeseer, and Pubmed dataset.

## C.6 Robustness against Adaptive Attack

We further investigate the capability of SG-GSR against adaptive attacks, since existing defense algorithms are shown to be more vulnerable to the adaptive attacks than the transferred attacks generated using the GCN surrogate model [27]. We tried our best to implement the adaptive attack on state-of-the-art models, i.e., RGCN [47], ProGNN [12], RSGNN [4] and STABLE [17], and used them as our baselines. The meta attack [51] is used to generate each adaptive attack. In the following, we describe how we implemented the adaptive attack on each model:

For **RGCN** and **ProGNN**, we implemented the adaptive attack following the same setting used in [27].

Since **STABLE** is a two-stage method, which employs an unsupervised model and a GCN, we had to modify the metattack algorithm to generate the adaptive attack for STABLE. Specifically, STABLE first learns the node embeddings using the DGI[6] backbone, and obtains the refined graph using kNN algorithm based on the node embeddings. Second, it trains GCN with the refined graph. Hence, we implemented the adaptive attack for STABLE by using the GCN model as the surrogate model at the second stage, where the input graph is a refined graph obtained at the first stage.

For **RSGNN**, the implementation of adaptive attacks suffered from the memory issue due to the design of metattack, which utilizes the training procedure of the surrogate model. In other words, its memory demand surpassed 48GB on a GPU. Hence, we implemented adaptive attacks on RSGNN by reducing the number of

inner-training iterations to a level that enables execution without a memory issue, i.e., 10 inner-training iterations. However, we discovered that the reduced number of inner-training iterations generates relatively weak attacks due to the underfitting of the surrogate model. Specifically, considering that RSGNN is trained for 1,000 iterations in the official source code, reducing the number of inner-training iterations to 10 results in a severe underfitting, thereby generating weak attacks. Note that adaptive attack with 10 inner-training iterations is even less effective than the + *Meta 25%* as shown in Table 14. To further vadliate that reducing the number of inner-training iterations indeed results in generating weak attacks, we also report the result of RSGNN with only 1 inner-training iteration. In Table 14, we clearly observe that as the number of iterations is further reduced from 10 to 1, the performance of RSGNN further improves, indicating that even weaker attacks are generated.

Instead of such weaker attacks, we conducted another experiment for RSGNN. While doing the above experiments, we discovered that it is the graph structure learner (GSL) component that causes the memory issue (Please note that RSGNN consists of two core components, i.e., a graph structure learner (GSL) and a GNN classifier). Hence, as an alternative approach, we first trained the GSL component until convergence to obtain the best structure, and then used the GNN classifier as the surrogate model to generate adaptive attacks, where the input to the surrogate model is the refined graph obtained from the GSL.

In Table 15, we observe that the adaptive attacks indeed degrade the performance of the victim models more significantly than the metattack, which is a non-adaptive attack, as shown in [27]. Although the performance of SG-GSR has also deteriorated, SG-GSR still outperforms other baselines on the adaptive attacks, implying that SG-GSR effectively refines the graph structure, and the learned representations are robust to the adaptive attacks.

The robustness of SG-GSR under adaptive attacks can be attributed to its ability to utilize an extracted clean sub-graph. While adaptive attacks can significantly perturb the graph structure to degrade the models, SG-GSR responds by extracting a clean sub-graph from the attacked graph. We argue that our results, derived from various attack methods such as metattack, nettack, injection

**Table 14: Node classification performance comparison under adaptive attacks on RSGNN with varying inner-train iterations.**

| Dataset | Setting | RSGNN |
|---------|---------|-------|
| Cora | + Meta 25% | 81.8±0.3 |
| | + Adaptive attack 10 inner-train iters | 82.8±0.4 |
| | + Adaptive attack 1 inner-train iters | 83.0±0.9 |
| Citeseer | + Meta 25% | 73.9±0.7 |
| | + Adaptive attack 10 inner-train iters | 75.8±0.2 |
| | + Adaptive attack 1 inner-train iters | 76.4±0.3 |

**Table 15: Node classification performance under adaptive attacks.**

| Dataset | Setting | RGCN | ProGNN | RSGNN | STABLE | SG-GSR |
|---------|---------|------|--------|-------|--------|--------|
| Cora | Clean | 84.0±0.1 | 82.9±0.3 | 85.1±0.3 | 85.1±0.3 | **85.5±0.1** |
| | + Meta 25% | 53.4±0.3 | 70.7±0.2 | 81.8±0.3 | 79.0±0.4 | **83.1±0.5** |
| | + Adaptive attack | 52.9±0.4 | 56.6±0.0 | 75.5±1.1 | 76.6±0.6 | **81.8±0.2** |
| Citeseer | Clean | 73.0±0.4 | 72.5±0.5 | 74.4±1.1 | **75.5±0.7** | 75.4±0.2 |
| | + Meta 25% | 58.6±0.9 | 68.4±0.6 | 73.9±0.7 | 73.4±0.3 | **75.2±0.1** |
| | + Adaptive attack | 40.0±0.2 | 68.0±0.0 | 70.8±0.8 | 71.1±0.7 | **74.3±0.6** |

attack, and adaptive attack, demonstrate that the strategy of extracting a clean sub-graph is a potent and effective defense against a range of attack methods.

In addition to the adaptive attack, we conduct the experiments using a bare minimum robustness unit test suggested in [27] (Please refer to line 6 in Section 6 of [27]). That is, we choose four strongest attacks that are transferred from ProGNN, RGCN, GRAND, and Softmedian GDC models. In Table 16, SG-GSR generally outperforms the baselines under these four attacks. From these results, we again demonstrate the efficacy of SG-GSR against adaptive attacks.

## C.7 Application on Other Downstream Tasks.

Although SG-GSR mainly focus on the node classification task, the graph structure refinement (GSR) method in SG-GSR can be applied to other downstream tasks such as link prediction and node clustering. We compared the performance of SG-GSR with the current state-of-art GSR baselines, i.e., RSGNN and STABLE, on link prediction and node clustering. Although both baselines and SG-GSR are primarily tailored for node classification, they can be adapted for the link prediction and node clustering task with minor modifications.

*C.7.1 Link prediction results.* Considering the link prediction task, RSGNN incorporates a link predictor that infers the latent graph structure using the embeddings of paired nodes. Based on the node embedding matrix, RSGNN computes the dot-product between a pair of node representations to calculate the likelihood of a link. STABLE is a 2-step GSR method that 1) learns node representations in an unsupervised manner using the DGI framework, and 2) constructs a $k$-NN graph as a refined graph structure using the fixed node representations, which is followed by a GCN classifier. Hence, we compute the dot-product between a pair of node representations from step 1 to calculate the likelihood of a link. Similar to RSGNN, SG-GSR also has a link predictor in the

GSR module, which can be used for link prediction. Note that the attention coefficient between node $i$ and node $j$ is computed as $e_{ij}^{l+1} = [(\mathbf{W}^{l+1}\mathbf{h}_i^l)^\top \cdot \mathbf{W}^{l+1}\mathbf{h}_j^l]/\sqrt{F^{l+1}}$, which we use it as the likelihood of a link between node $i$ and node $j$.

For the evaluation protocol, we split the given edges into training and test edges, in a 5:5 ratio [1]. Only the training edges are shown to the link predictor, and we evaluate the models on test edges. Specifically, we use the node representations obtained by the trained models to produce the link score of the test edges. In Table 17, we observe that SG-GSR outperforms RSGNN and STABLE in the link prediction task in terms of AUROC. We attribute this to the fact that our proposed sub-graph extraction module effectively finds the clean edges from the attacked structure and proposed graph augmentation module successfully supplements the structural information that reflects the real-world graph properties, which leads GSR to accurately predict reliable links.

*C.7.2 Node clustering results.* Generally, the node clustering task is performed to confirm the quality of learned node representations under the unsupervised learning setting. However, as RSGNN, STABLE, and SG-GSR are designed for the node classification task under the supervised setting, we use the node embeddings obtained from the intermediate GNN layer, which is followed by the classification layer to run the k-means algorithm, and obtain the cluster assignments of nodes. In Table 18, we observe that SG-GSR outperforms RSGNN and STABLE in the node clustering task in terms of NMI, which indicates that SG-GSR effectively acquires more class separable node embeddings. We attribute it to the fact that SG-GSR minimizes the effect of malicious inter-class edges that incur the vague class boundary in the representation space. Regarding the ability of SG-GSR that removes the inter-class edges, refer to Fig. 15 in Appendix C.4.

## C.8 Fair Comparison of Backbone Model

Given that the baselines employ GCN as their backbone model, while SG-GSR chooses SuperGAT, we ensure fair comparisons by using SuperGAT as the backbone for the baselines. We reimplement two baselines SLAPS and STABLE using SuperGAT as the backbone model. It is important to note that the remaining baselines can not adopt SuperGAT as the backbone due to the following reasons:

- The training procedure of ProGNN is similar to that of SuperGAT, as the structure learner of ProGNN utilizes the edge weight to propagate messages, where the edge weight is forced to be close to the original adjacency matrix. This is equivalent to the link prediction loss in SuperGAT.
- The training procedure of RSGNN is similar to that of SuperGAT, as it utilizes the edge weight to propagate messages, where the edge weight is assigned by the link predictor.
- RGCN, ELASTIC, AirGNN, and EvenNet each employ their own distinct message-passing mechanism. Therefore, it is not logical to adopt SuperGAT as the backbone of these models.

In Table 19, we observed that when SuperGAT backbone is adopted, the performance of baselines are generally enhanced. However, SG-GSR still outperforms the baselines adopting SuperGAT. The reason is that the fundamental problems of feature-based and

**Table 16: Node classification accuracy on a bare minimum robustness unit test.**

| Dataset | Transferred from | SuperGAT | RGCN | ProGNN | GEN | ELASTIC | AirGNN | SLAPS | RSGNN | CoGSL | STABLE | EvenNet | SE-GSL | SG-GSR |
|---|---|---|---|---|---|---|---|---|---|---|---|---|---|---|
| Cora-ML | ProGNN | 61.1±0.3 | 50.4±0.4 | 73.1±0 | 73.2±0 | 71±0.8 | 56.4±0.7 | 72.6±0.5 | **82.2±0.5** | 50.1±0 | 74.4±1 | 72.7±0.6 | 70.2±0.7 | 81.4±0.4 |
|  | RGCN | 69.7±0.4 | 52.9±0.4 | 76.3±0 | 65.7±0 | 72.4±1 | 50.5±0.4 | 72.6±0.5 | 79.6±0.2 | 51.3±0 | 76.7±1.4 | 76.1±0.4 | 65.5±0.3 | **81.6±0.6** |
|  | GRAND | 47.6±3.1 | 42.8±0.7 | 62.8±0 | 70.5±0 | 68±0.4 | 51.8±0.8 | 72.6±0.5 | 81.3±0.2 | 44.7±0 | 77.7±0.6 | 70.9±0 | 62.6±0.1 | **82.4±0.7** |
|  | Soft-Median GDC | 74.3±1.3 | 66.6±0.1 | 78.9±0 | 78.3±0 | 77.4±0.4 | 68.2±0.8 | 72.6±0.5 | 82.3±0.2 | 54.5±0 | 78.3±1.1 | 78.9±0.3 | 67.0±0.7 | **84.1±0.6** |
| Citeseer | ProGNN | 54.4±2.3 | 43.3±0.3 | 64.2±0 | 67±0 | 65.8±1.6 | 47.4±0.4 | 73.5±0.2 | **75.6±0.6** | 44.6±0 | 70.9±2 | 68.5±0.8 | 53.3±0.7 | 74.5±0.3 |
|  | RGCN | 67±1.3 | 40±0.2 | 57.9±0 | 65.1±0 | 66.7±0.1 | 37.4±1.1 | 73.5±0.2 | 73.9±0.2 | 49.6±0 | 72.5±0.3 | 71.1±0.8 | 63.7±0.6 | **74.1±0.7** |
|  | GRAND | 50.3±0.1 | 44.1±0.2 | 60±0 | 70.4±0 | 65.8±1.6 | 48.2±0.8 | 73.5±0.2 | 75.4±0.8 | 51.8±0 | 71.7±0.3 | 67.1±1.7 | 59.8±0.8 | **75.8±0.4** |
|  | Soft-Median GDC | 54.8±0.7 | 47.9±0.1 | 63.9±0 | 68.5±0 | 66.6±1.5 | 50.9±0.4 | 73.5±0.2 | **76.7±0.4** | 48.1±0 | 72.5±0.4 | 68.7±0.1 | 60.0±0.5 | 75.7±0.5 |

**Table 17: Link prediction performance under non-targeted attack (i.e., *metattack*) and feature attack.**

| Dataset | Setting | RSGNN | STABLE | SG-GSR |
|---|---|---|---|---|
| Cora | Clean | 89.3±0.1 | 93.1±0.1 | **94.7±0.7** |
|  | + Meta 25% | 86.9±0.0 | 91.5±0.1 | **93.8±0.3** |
|  | + Feat attack | 82.6±0.1 | 72.5±3.8 | **89.8±0.6** |
| Citeseer | Clean | 90.1±0.5 | **96.5±0.1** | 96.3±1.0 |
|  | + Meta 25% | 88.0±0.2 | 95.8±0.2 | **96.2±0.3** |
|  | + Feat attack | 83.4±0.2 | 84.6±0.4 | **93.9±0.2** |

**Table 18: Node clustering performance under non-targeted attack (i.e., *metattack*) and feature attack.**

| Dataset | Setting | RSGNN | STABLE | SG-GSR |
|---|---|---|---|---|
| Cora | Clean | 62.9±1.0 | 64.5±1.2 | **65.8±1.6** |
|  | + Meta 25% | 57.8±2.2 | 54.2±0.6 | **60.7±0.5** |
|  | + Feat attack | 43.0±2.6 | 32.2±3.0 | **44.3±0.5** |
| Citeseer | Clean | 49.6±0.7 | 47.5±1.3 | **50.1±0.4** |
|  | + Meta 25% | 42.5±5.1 | 44.1±0.7 | **47.4±1.0** |
|  | + Feat attack | 35.1±0.4 | 25.3±1.1 | **35.6±1.0** |

**Table 19: Comparison with baselines when SuperGAT backbone is used.**

| Dataset | Setting | SLAPS-SuperGAT | STABLE-SuperGAT | SG-GSR |
|---|---|---|---|---|
| Cora | Clean | 74.3±0.1 | 83.5±1.2 | **85.5±0.1** |
|  | + Meta 25% | 73.7±0.3 | 75.1±0.5 | **83.1±0.5** |
|  | + Feat attack | 50.6±0.4 | 50.5±1.3 | **67.6±1.4** |
| Citeseer | Clean | 74.1±0.4 | 74.8±0.2 | **75.4±0.2** |
|  | + Meta 25% | 74.0±0.6 | 73.6±0.6 | **75.2±0.1** |
|  | + Feat attack | 58.6±0.4 | 57.5±1.0 | **66.8±1.0** |

multi-faceted GSR (i.e., assuming clean node features and moderate structural attacks) cannot be resolved by simply replacing the backbone network with SuperGAT.

## C.9 Further Comparison with GCN-SVD and GARNET

We further compare SG-GSR with some related GSR methods such as GCN-SVD [7] and GARNET [6]. In Table 20, we observe that SG-GSR outperforms them with various backbone networks. From the results, we demonstrate that our proposed graph refinement strategy is more robust compared with GCN-SVD and GARNET.

**Table 20: Comparison with baselines when SuperGAT backbone is used.**

| Dataset | Setting | GCN-SVD | GARNET GCN | GARNET GRPGNN | GARNET SuperGAT | SG-GSR |
|---|---|---|---|---|---|---|
| Cora | Clean | 77.8±0.1 | 81.9±0.3 | 83.2±0.4 | 79.2±1.1 | **85.5±0.1** |
|  | '+ Meta 25% | 55.8±1.8 | 74.8±1.3 | 78.9±0.9 | 56.9±2.0 | **83.1±0.5** |
|  | '+ Feat attack | 50.6±0.4 | 60.7±1.0 | 62.6±1.2 | 54.9±1.7 | **67.6±1.4** |
| Citeseer | Clean | 69.7±0.5 | 72.6±0.5 | 75.1±0.6 | 69.7±0.6 | **75.4±0.2** |
|  | '+ Meta 25% | 61.4±0.8 | 67.8±0.6 | 72.4±0.9 | 67.4±1.1 | **75.2±0.1** |
|  | '+ Feat attack | 49.0±0.8 | 56.8±0.9 | 58.0±1.3 | 34.2±3.9 | **66.8±1** |
| Pubmed | Clean | 84.4±0.1 | 86.2±0.3 | 86.8±0.2 | OOM | **87.6±0.2** |
|  | '+ Meta 25% | 76.3±0.6 | 86.2±0.2 | 86.7±0.1 | OOM | **87.3±0.2** |
|  | '+ Feat attack | 59.3±0.3 | 52.8±0.8 | 62.3±0.6 | OOM | **65.5±0.5** |

# D DETAILS ON EXPERIMENTAL SETTINGS

## D.1 Datasets

We evaluate SG-GSR and baselines on **five existing datasets** (i.e., Cora [12], Citeseer [12], Pubmed [12], Polblogs [12], and Amazon [28]) and **two newly introduced datasets** (i.e., Garden and Pet) that are proposed in this work based on Amazon review data [10, 26] to mimic e-commerce fraud (Refer to Appendix D.2 for details). The statistics of the datasets are given in Table 21. Note that since there are do not exist node feature matrix in Polblogs dataset, we use the identity matrix as the node features, following the setting of existing work [12]. For each graph, we use a random 1:1:8 split for training, validation, and testing. These seven datasets can be found in these URLs:

- **Cora**: https://github.com/ChandlerBang/Pro-GNN
- **Citeseer**: https://github.com/ChandlerBang/Pro-GNN
- **Pubmed**: https://github.com/ChandlerBang/Pro-GNN
- **Polblogs**: https://github.com/ChandlerBang/Pro-GNN
- **Amazon**: https://pytorch-geometric.readthedocs.io/en/latest/
- **Garden**: http://jmcauley.ucsd.edu/data/amazon/links.html
- **Pet**: http://jmcauley.ucsd.edu/data/amazon/links.html

**Table 21: Statistics for datasets.**

| Domain | Dataset | # Nodes | # Edges | # Features | # Classes |
|---|---|---|---|---|---|
| Citation graph | Cora | 2,485 | 5,069 | 1,433 | 7 |
|  | Citeseer | 2,110 | 3,668 | 3,703 | 6 |
|  | Pubmed | 19,717 | 44,338 | 500 | 3 |
| Blog graph | Polblogs | 1,222 | 16,714 | / | 2 |
| Co-purchase graph | Amazon | 13,752 | 245,861 | 767 | 10 |
| Co-review graph | Garden | 7,902 | 19,383 | 300 | 5 |
|  | Pet | 8,664 | 69,603 | 300 | 5 |

## D.2 Data generation process on e-commerce fraud.

In this work, we newly design and publish two novel graph benchmark datasets, i.e., Garden and Pet, that simulate real-world fraudsters' attacks on e-commerce systems. To construct a graph, we use the metadata and product review data of two categories, "Patio, Lawn and Garden" and "Pet Supplies," in Amazon product review data [10, 26]. Specifically, we generate a clean product-product graph, where the node feature is bag-of-words representation of product reviews, the edges indicate the co-review relationship between two products reviewed by the same user, and the node label is the product category. For the attacked graph, we imitate the behavior of fraudsters/attackers in a real-world e-commerce platform. As the attackers interact with randomly chosen products (i.e., write fake product reviews), not only numerous malicious co-review edges are added to the graph structure, but also noisy random reviews (i.e., noisy bag-of-words representations) are injected into the node features. More precisely, we set the number of fraudsters to 100, and moreover, the number of reviews written by each fraudster is set to 100 in the Garden dataset and to 200 in the Pet dataset. To generate a fake review text, we randomly select a text from existing reviews and copy it to the products that are under attack. This method ensures that the fake reviews closely resemble the style and content of real reviews while also containing irrelevant content that makes it more challenging to predict the product category. The data generation code is also available at https://anonymous.4open.science/r/torch-SG-GSR-97CC

We again emphasize that while existing works primarily focus on artificially generated attack datasets, to the best of our knowledge, this is the first work proposing new graph benchmark datasets for evaluating the robustness of GNNs under adversarial attacks that closely imitate a real-world e-commerce system containing malicious fraudsters. We expect these datasets to foster practical research in adversarial attacks on GNNs.

## D.3 Baselines

We compare SG-GSR with a wide range of GNN methods designed to defend against structural attacks, which includes robust node representations methods (i.e., RGCN [47], ELASTIC [23], and EvenNet [16]), feature-based GSR (i.e., ProGNN [12], SLAPS [8], and RSGNN [4]), multi-faceted GSR (i.e., SuperGAT [13], GEN [35], CoGSL [20], OAGS [29], STABLE [17], and SE-GSL [48]). We also consider AirGNN [22] which is designed to defend against the feature attack/noise.

The publicly available implementations of baselines can be found at the following URLs:

- **SuperGAT** [13] : https://github.com/dongkwan-kim/SuperGAT
- **RGCN** [47] : https://github.com/DSE-MSU/DeepRobust
- **ProGNN** [12] : https://github.com/ChandlerBang/Pro-GNN
- **GEN** [35] : https://github.com/BUPT-GAMMA/Graph-Structure-Estimation-Neural-Networks
- **ELASTIC** [23] : https://github.com/lxiaorui/ElasticGNN
- **AirGNN** [22] : https://github.com/lxiaorui/AirGNN
- **SLAPS** [8] : https://github.com/BorealisAI/SLAPS-GNN
- **RSGNN** [4] : https://github.com/EnyanDai/RSGNN
- **CoGSL** [20] : https://github.com/liun-online/CoGSL
- **STABLE** [17] : https://github.com/likuanppd/STABLE

- **EvenNet** [16] : https://github.com/Leirunlin/EvenNet
- **SE-GSL** [48] : https://github.com/ringbdstack/se-gsl
- **OAGS** [29] : As there is no publicly available implementation of OAGS, we tried our best to implement OAGS ourselves. However, we failed to reproduce the presented results due to the lack of implementation details in its main paper. Specifically, there is no detailed derivation of $D_{KL}(q_\phi(\hat{\mathbf{A}})||p(\hat{\mathbf{A}}))$ in Eq. 19 of its main paper, where $\hat{\mathbf{A}}$ is the estimated graph structure, $p(\hat{\mathbf{A}}_{ij}) \sim \mathcal{N}(\dot{\mathbf{A}}, 0)$ is the random prior over $\hat{\mathbf{A}}$ given the observed graph $\mathbf{A}$, and $q_\phi(\hat{\mathbf{A}})$ is the approximate posterior with free parameters $\mu_{ij}^\phi$ and $\sigma_{ij}^\phi$. Moreover, $\dot{\mathbf{A}} = \theta_1 \mathbf{A} + \theta_2(1 - \mathbf{A})$ is empirically set with hyperparameters $\theta_1$ and $\theta_2$. Please note that implementing the term $D_{KL}(q_\phi(\hat{\mathbf{A}})||p(\hat{\mathbf{A}}))$ is important because this is the loss directly related to estimating the graph structure. Hence, we fix the estimated graph structure as the mean of the prior $\dot{\mathbf{A}} = \theta_1 \mathbf{A} + \theta_2(1 - \mathbf{A})$ and implement the other parts. We report our implemented results in Table 22.

**Table 22: Node classification performance under non-targeted attack (i.e., *metattack*) and feature attack.**

| Dataset | Setting | OAGS | SG-GSR |
|---------|---------|------|--------|
| Cora | Clean | 67.27±3.89 | **85.48±0.05** |
| | + Meta 25% | 44.20±1.13 | **83.10±0.47** |
| | + Feat attack | 44.15±1.95 | **67.56±1.40** |
| Citeseer | Clean | 61.22±4.10 | **75.36±0.21** |
| | + Meta 25% | 51.74±1.00 | **75.22±0.10** |
| | + Feat attack | 47.06±1.01 | **66.82±1.02** |
| Pubmed | Clean | 50.12±1.18 | **87.55±0.22** |
| | + Meta 25% | 50.19±1.77 | **87.27±0.19** |
| | + Feat attack | 49.85±1.66 | **65.49±0.49** |
| Polblogs | Clean | 93.97±0.08 | **96.22±0.08** |
| | + Meta 25% | 85.38±0.58 | **87.80±0.72** |
| Amazon | Clean | 56.03±1.43 | **91.06±0.17** |
| | + Meta 25% | 54.85±0.85 | **89.23±0.24** |
| | + Feat attack | 53.48±1.03 | **87.21±0.39** |

## D.4 Evaluation Protocol

We compare SG-GSR and the baselines under poisoning structure and feature attacks, following existing robust GNN works [4, 12, 16, 17, 20]. We consider three attack scenarios, i.e., structure attacks, structure-feature attacks, and e-commerce fraud. Note that in this work we mainly focus on graph modification attacks (i.e., modifying existing topology or node features). For structure attacks, we adopt *metattack* [51] and *nettack* [50] as a non-targeted and targeted attack method, respectively, which are the commonly used attacks in existing defense works [12, 17]. For structure-feature attacks, we further inject independent random Gaussian noise into the node features as in [20, 22, 29]. More specifically, we add a noise vector $\gamma \cdot \mathbf{m}_i^{\text{noise}} \in \mathbb{R}^F$ to the node $i$, where $\gamma$ is set to 0.5, which is a noise ratio, and each element of $\mathbf{m}_i^{\text{noise}}$ is independently sampled from the standard normal distribution. Note that we only add the noise vector to a subset of the nodes (i.e., 50%) since it is more realistic that only certain nodes are attacked/noisy rather than all of them. Lastly, we introduce two new benchmark datasets for attacks that mimic e-commerce fraud (Refer to Sec. D.2).

**Table 25: Hyperparameter settings on SG-GSR for Table 3.**

| Dataset | Setting | lr | dropout | $\lambda_{\text{sp}}$ | $\lambda_{\text{fs}}$ | $\lambda_{\text{aug}}$ | $\lambda_{\mathcal{E}}$ |
|---------|---------|-----|---------|------|------|------|-----|
| Garden | Clean | 0.01 | 0.4 | 1 | 0.9 | 0.1 | 0.5 |
|        | + Fraud | 0.001 | 0.2 | 0.9 | 0.9 | 0.7 | 2.0 |
| Pet | Clean | 0.005 | 0.0 | 0.7 | 0.9 | 0.3 | 0.5 |
|     | + Fraud | 0.05 | 0.2 | 0.5 | 0.9 | 0.5 | 2.0 |

---

**Algorithm 1** Training Algorithm of SG-GSR

---

**Input**: Graph $\mathcal{G} = \langle \mathcal{V}, \mathcal{E}, \mathbf{X} \rangle$, Initial parameters $\{\mathbf{W}\}_{l=1}^{L}$

/* Phase 1 */

Pretrain node2vec on $\mathcal{G}$ to obtain the node embeddings $\mathbf{H}^{\text{sp}}$

Calculate structural proximity $S_{ij}^{\text{sp}}$ and feature similarity $S_{ij}^{\text{fs}}$ for $(i, j) \in \mathcal{E}$

Generate two $k$-NN graphs $\tilde{\mathcal{E}}_k^{\text{FS}}$ and $\tilde{\mathcal{E}}_k^{\text{SP}}$ from $\mathbf{H}^{\text{sp}}$ and $\mathbf{X}$

Extract a clean sub-graph $\tilde{\mathcal{G}} = \langle \tilde{\mathcal{V}}, \tilde{\mathcal{E}}, \tilde{\mathbf{X}} \rangle$ from $\mathcal{G}$ using $S_{ij}^{\text{sp}}, S_{ij}^{\text{fs}}$

/* Phase 2 */

**for all** epoch **do**

    Obtain $\tilde{\mathcal{E}}^{\text{aug}}$ by Eqn. 5

    Calculate the node representations $\{\mathbf{h}_i^l\}_{l=1}^{L}$ based on $\tilde{\mathcal{E}}^{\text{aug}}$

    Split $\tilde{\mathcal{E}}^{\text{aug}}$ into $\tilde{\mathcal{E}}_{\text{LL}}^{\text{aug}}$, $\tilde{\mathcal{E}}_{\text{HL}}^{\text{aug}}$, and $\tilde{\mathcal{E}}_{\text{HH}}^{\text{aug}}$

    Calculate $L_{\tilde{\mathcal{E}}_{\text{LL}}^{\text{aug}}}^{l}$, $L_{\tilde{\mathcal{E}}_{\text{HL}}^{\text{aug}}}^{l}$, and $L_{\tilde{\mathcal{E}}_{\text{HH}}^{\text{aug}}}^{l}$ via Eqn. 2

    Calculate $L_{\text{final}} = L_{\tilde{\mathcal{V}}} + \lambda_{\mathcal{E}} \sum_{l=1}^{L} (L_{\tilde{\mathcal{E}}_{\text{LL}}^{\text{aug}}}^{l} + L_{\tilde{\mathcal{E}}_{\text{HL}}^{\text{aug}}}^{l} + L_{\tilde{\mathcal{E}}_{\text{HH}}^{\text{aug}}}^{l})$

    Update parameters $\{\mathbf{W}_{l=1}^{L}\}$ to minimize $L_{\text{final}}$.

**end for**

**return:** model parameter $\{\mathbf{W}\}_{l=1}^{L}$

---

## D.5 Implementation Details

For each experiment, we report the average performance of 3 runs with standard deviations. For all baselines, we follow the implementation details presented in their original papers.

For SG-GSR, the learning rate and dropout are tuned from {0.05, 0.01, 0.005, 0.001} and {0.0, 0.2, 0.4, 0.6, 0.8}, respectively, and weight decay is fixed to 0.0005. For the GSR module, we fix the number of GNN layers, hidden units, and attention heads as 2, 16, and 8, respectively. When calculating the link predictor loss $L_{\mathcal{E}}^{l}$, we use the arbitrarily selected negative samples $\mathcal{E}^-$, the size of which is set to $p_n \cdot |\mathcal{E}|$ where the negative sampling ratio $p_n \in \mathbb{R}^+$ is set to 0.5 in Cora, Citeseer, and Polblogs, and to 0.25 in Pubmed, Amazon, Garden, and Pet datasets. And we tune a coefficient $\lambda_{\mathcal{E}}$ for the link predictor loss from {0.2, 0.5, 1, 2, 3, 4, 5}.

For the clean sub-graph extraction module, $\lambda_{\text{sp}}$ and $\lambda_{\text{fs}}$ are tuned from {1.0, 0.9, 0.7, 0.5, 0.3 }. For the graph augmentation, the $k$ value in $\tilde{\mathcal{E}}_k^*$ is set to 5 in Cora, Citeseer, Pubmed, and Garden, to 10 in Pet, to 50 in Polblogs, and to 30 in Amazon. And the $\lambda_{\text{aug}}$ is tuned from {0.1, 0.3, 0.5, 0.7, 0.9}, For group-training strategy, we split the edge set in a more fine-grained way, i.e., $\tilde{\mathcal{E}}_{\text{LL}}^{\text{aug}}$, $\tilde{\mathcal{E}}_{\text{MM}}^{\text{aug}}$, $\tilde{\mathcal{E}}_{\text{HH}}^{\text{aug}}$, $\tilde{\mathcal{E}}_{\text{ML}}^{\text{aug}}$, $\tilde{\mathcal{E}}_{\text{HL}}^{\text{aug}}$, and $\tilde{\mathcal{E}}_{\text{HM}}^{\text{aug}}$, where L, M, and H indicate low-, mid-, and high-degree nodes. We report the details of hyperparameter settings in Table 23, 24, and 25.

**Table 23: Hyperparameter settings on SG-GSR for Table 1.**

| Dataset | Setting | lr | dropout | $\lambda_{\text{sp}}$ | $\lambda_{\text{fs}}$ | $\lambda_{\text{aug}}$ | $\lambda_{\mathcal{E}}$ |
|---------|---------|-----|---------|------|------|------|-----|
| Cora | Clean | 0.005 | 0.6 | 1.0 | 1.0 | 0.5 | 2.0 |
|      | + Meta 25% | 0.01 | 0.6 | 1.0 | 0.5 | 0.9 | 3.0 |
|      | + Feat attack | 0.01 | 0.4 | 1.0 | 0.7 | 0.9 | 3.0 |
| Citeseer | Clean | 0.001 | 0.6 | 0.9 | 0.9 | 0.3 | 1.0 |
|          | + Meta 25% | 0.001 | 0.6 | 1.0 | 0.5 | 0.3 | 3.0 |
|          | + Feat attack | 0.005 | 0.6 | 1.0 | 0.7 | 0.7 | 5.0 |
| Pubmed | Clean | 0.05 | 0.2 | 1.0 | 0.9 | 0.7 | 4.0 |
|        | + Meta 25% | 0.05 | 0.2 | 0.9 | 0.7 | 0.9 | 2.0 |
|        | + Feat attack | 0.01 | 0.0 | 0.5 | 1.0 | 0.9 | 4.0 |
| Polblogs | Clean | 0.01 | 0.2 | 1.0 | 1.0 | 0.5 | 3.0 |
|          | + Meta 25% | 0.05 | 0.8 | 0.3 | 1.0 | 0.9 | 3.0 |
| Amazon | Clean | 0.005 | 0.2 | 0.5 | 1.0 | 0.3 | 0.5 |
|        | + Meta 25% | 0.005 | 0.2 | 0.5 | 1.0 | 0.7 | 0.5 |
|        | + Feat attack | 0.005 | 0.2 | 0.7 | 0.7 | 0.7 | 0.5 |

**Table 24: Hyperparameter settings on SG-GSR for Table 2.**

| Dataset | Setting | lr | dropout | $\lambda_{\text{sp}}$ | $\lambda_{\text{fs}}$ | $\lambda_{\text{aug}}$ | $\lambda_{\mathcal{E}}$ |
|---------|---------|-----|---------|------|------|------|-----|
| Cora | Clean | 0.005 | 0.2 | 1.0 | 0.9 | 0.5 | 0.5 |
|      | + Net 5 | 0.001 | 0.0 | 0.9 | 0.5 | 0.7 | 4.0 |
|      | + Feat attack | 0.01 | 0.4 | 0.7 | 0.0 | 0.3 | 5.0 |
| Citeseer | Clean | 0.005 | 0.4 | 0.7 | 0.5 | 0.9 | 2.0 |
|          | + Net 5 | 0.01 | 0.6 | 1.0 | 0.5 | 0.1 | 5.0 |
|          | + Feat attack | 0.005 | 0.4 | 0.7 | 0.7 | 0.3 | 3.0 |
| Pubmed | Clean | 0.05 | 0.0 | 0.5 | 1.0 | 0.5 | 4.0 |
|        | + Net 5 | 0.05 | 0.2 | 0.9 | 0.7 | 0.3 | 1.0 |
|        | + Feat attack | 0.05 | 0.0 | 0.7 | 1.0 | 0.5 | 0.5 |
| Polblogs | Clean | 0.01 | 0.8 | 0.7 | 1 | 0.9 | 0.2 |
|          | + Net 5 | 0.005 | 0.8 | 0.5 | 1 | 0.1 | 0.5 |



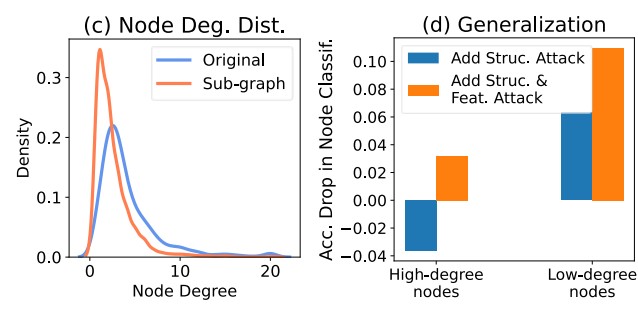

Figure 16: (a) Clean rate of the extracted edges and remaining edges over the ratio of extracted edges. (b) Node classification accuracy under structure attack and clean rate of extracted edges over the ratio of extracted edges. (c) Node degree distribution of original graph and extracted sub-graph. (d) Accuracy drop in node classification under attacks for high/low-degree nodes. Citeseer dataset is used. *Struc. Attack* indicates *metattack* 25% and *Feat. Attack* indicates Random Gaussian noise 50%.

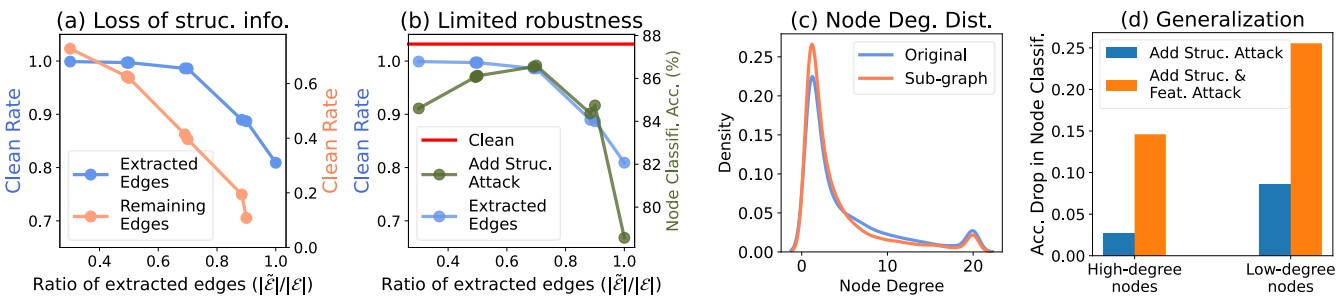

Figure 17: (a) Clean rate of the extracted edges and remaining edges over the ratio of extracted edges. (b) Node classification accuracy under structure attack and clean rate of extracted edges over the ratio of extracted edges. (c) Node degree distribution of original graph and extracted sub-graph. (d) Accuracy drop in node classification under attacks for high/low-degree nodes. Pubmed dataset is used. *Struc. Attack* indicates *metattack* 25% and *Feat. Attack* indicates Random Gaussian noise 50%.

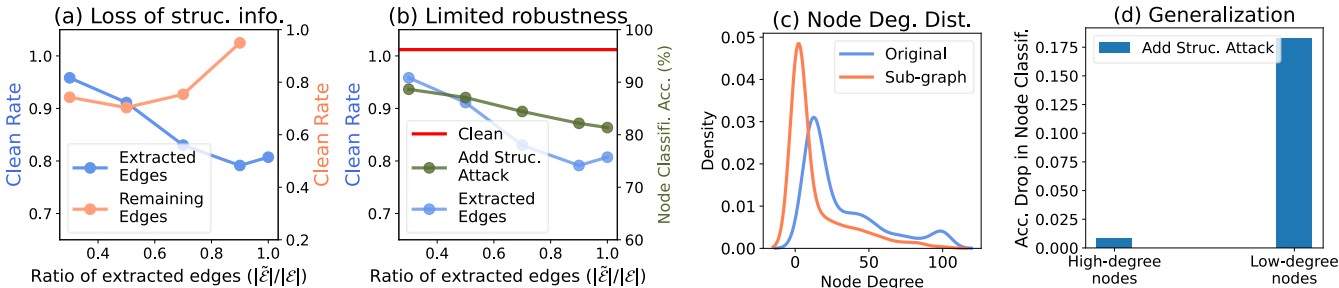

Figure 18: (a) Clean rate of the extracted edges and remaining edges over the ratio of extracted edges. (b) Node classification accuracy under structure attack and clean rate of extracted edges over the ratio of extracted edges. (c) Node degree distribution of original graph and extracted sub-graph. (d) Accuracy drop in node classification under attacks for high/low-degree nodes. Polblogs dataset is used. *Struc. Attack* indicates *metattack* 25%.

