# OpenReview forum: "Self-guided Robust Graph Structure Refinement"
_ACM.org/TheWebConf/2024/Conference — TheWebConf24 Oral_

### Official Review · Reviewer_iBFg · 2023-11-12

**Novelty:** 5
**Technical Quality:** 6

**Review:**

Existing GSR methods are limited by narrow assumptions, resulting in restricted applicability in real-world scenarios. To solve this problem, this paper proposes a self-guided GSR framework  (SG-GSR), which utilizes a clean sub-graph found within the given attacked graph itself. Furthermore, it proposes a novel graph augmentation and a group-training strategy to handle the two technical challenges in the clean sub-graph extraction: 1) loss of structural information, and 2) imbalanced node degree distribution. Extensive experiments demonstrate the effectiveness of SG-GSR under various scenarios including non-targeted attacks, targeted attacks, feature attacks, e-commerce fraud, and noisy node labels.
Advantages:
(1)The structural logic of the article is very clear.
(2)The idea is novel.
(3)The paper is well-experimental.

Disadvantage:
(1) The article is obscure and difficult to understand.
(2) The motivation of the article is clear but not innovative enough.
(3) The language of this paper needs further polished.

**Questions:**

(1)The paper uses obscure text and symbols without formulas or figures, making it hard to follow.
(2)To better understand the method, please adjust Figure 6 Overall architecture into the method section.
(3)In Graph Augmentation, this paper measures the importance of each edge from three aspects: class homophily, feature smoothness, and structural proximity. Next, in Group-training Strategy, this paper divides the edges into three groups: two high-degree nodes and two low-degree nodes, between a high-degree node and a low-degree node. Is the classification of edges in the training strategy related to the enhancement of previous nodes? There is no relevant information transfer between them in the article.
(4) Are there any differences between feature smoothness and structural proximity? Because the paper states that both are computed by the cosine similarity between all node pairs in the sub-graph. Please use a formula to distinguish.
(5) The appendix has 10 pages, which is too long.

**Ethics Review Description:**

There is no ethics issues with this paper.

**Reviewer Confidence:**

3: The reviewer is confident but not certain that the evaluation is correct

**Scope:**

3: The work is somewhat relevant to the Web and to the track, and is of narrow interest to a sub-community

---

### Official Review · Reviewer_juTw · 2023-11-17

**Novelty:** 4
**Technical Quality:** 4

**Review:**

This paper aims to propose a graph neural network that is robust to adversarial attacks. The proposed method firstly extracts the clean subgraph from the entire graph, and addresses two challenges on the extraction: loss of structural information and imbalance node degree distribution. The experiments are conducted on seven real-world graphs with twelve baselines.

Pros:
1. The paper is easy to follow.
2. There are justifications for the raised challenges.
3. The experiments are comprehensive, including ablation studies and further analysis.

Cons:
1. The experiments are not done rigorously.

I have read the rebuttals.

**Questions:**

1. In Figure 1, what is the performance of the proposed method?

2. In line 407, how is the class prediction probability matrix computed? Does it mean that you have to train a model on $\tilde{\mathcal{E}}$ first?

3. My largest concern is about the experiments. The proposed method have many hyperparameters and the authors have done an exhausted search on them. However, the details of hyperparameter search for the baselines are missing, which make the results much less convincing. Has hyperparameter search done on the baselines?

**Reviewer Confidence:**

2: The reviewer is willing to defend the evaluation, but it is likely that the reviewer did not understand parts of the paper

**Scope:**

3: The work is somewhat relevant to the Web and to the track, and is of narrow interest to a sub-community

---

### Official Review · Reviewer_CCbX · 2023-11-26

**Novelty:** 6
**Technical Quality:** 5

**Review:**

This paper focuses on robust graph structure refinement against adversarial attacks of GNNs. To solve the limitations of existing methods brought by narrow assumptions such as clean node features, moderate structural attacks, and the availability of external clean graphs, the authors proposed a self-guided GSR framework. The proposed framework utilizes a clean sub-graph with graph augmentation and group-training strategies to improve the robustness of GNNs. Extensive experiments are conducted to demonstrate the effectiveness of the proposed method.

Strength:
1. The motivation of this paper is clear and well-supported.
2. The paper is well-organized and easy to follow.
3. Extensive experiments are conducted to demonstrate the effectiveness of the proposed method.

Weakness:
1. The proposed graph augmentation strategies considering class homophily, feature smoothness, and structural proximity are heuristic. It would be better if theoretical analysis or guarantees were provided, which could further improve the solid of the proposed method.
2. The formatting of the paper can be further improved, as the font size of many tables and images is too small.

**Questions:**

1. Since the augmentation strategies considering class homophily, feature smoothness, and structural proximity are heuristic, are there any theoretical analyses or deeper experimental discussions that could be provided to further support the generality of the proposed method?

**Ethics Review Description:**

No ethical issues

**Reviewer Confidence:**

3: The reviewer is confident but not certain that the evaluation is correct

**Scope:**

4: The work is relevant to the Web and to the track, and is of broad interest to the community

---

### Official Review · Reviewer_fSZq · 2023-12-02

**Novelty:** 5
**Technical Quality:** 5

**Review:**

To address the limitations of existing GSR methods, which rely on clean node features, moderate structure attack, and external clean graphs, the authors propose a new GSR approach, SG-GSR, which extracts clean subgraphs from the attacked graph to eliminate injected malicious structural information. Faced with two challenges in extracting clean subgraphs, namely incomplete structural information and an imbalanced node degree distribution, the authors introduce the Graph Augmentation and Group-training methods. The effectiveness of the proposed approach is validated through extensive empirical experiments to mitigate the issues mentioned in GSR. Additionally, the paper introduces two new datasets to assess the effectiveness of SG-GSR in the context of e-commerce fraud scenarios.
Pros:
It’s uplifting to see that the proposed method, SG-GSR, achieves high robustness to not only structure attack but feature attack as well, which is a distinctive trait in other defending methods.
The empirical analysis conducted is comprehensive, encompassing various tasks, notably featuring a novel e-commerce fraud scenario. Additionally, the study incorporates multiple baseline comparisons to provide a thorough evaluation framework. Supplementary experiments presented in the appendix offer further insights into the factors contributing to the observed performance improvements.
Cons:
The computational complexity remains a concern, given its requirement for graph embedding in the pre-training stage and node-to-node similarity computation in every training epoch. Although it’s claimed that time consumption is acceptable, addressing this concern would benefit from evidence demonstrating the scalability of SG-GSR to large-scale datasets. Alternatively, the authors could enhance the discussion by including comparative experiments on time consumption with other GSR models.
Some of the inferences in the paper are not entirely convincing to me. For example, in section 4.4.2, the reduction in I_{ratio}, which represents the ratio of the maximum node degree to the minimum node degree, is insufficient to substantiate the claim of a more balanced node distribution.
One issue in the experimental section is that the set adversarial attack rates are either 0% or 25%, lacking an assessment of the model's performance at intermediate attack rates. As demonstrated in many articles on similar topics, attack rates ranging from 5% to 20% are commonly encountered. Besides, in real-world scenarios, setting the adversarial attack budget at 25% is considered somehow unpractical.

**Questions:**

Is there a missing transition from H^{sp} to S^{sp} in line 313?
If cosine similarity is utilized for computing similarity in line 313, the graph augmentation of 'Property 3: Structural proximity' appears redundant, given that its impact is to incorporate edges with high S^{sp} scores into the subgraph. Could it be applicable to adjust \lambda_{sp} to attain a similar effect?

**Reviewer Confidence:**

3: The reviewer is confident but not certain that the evaluation is correct

**Scope:**

3: The work is somewhat relevant to the Web and to the track, and is of narrow interest to a sub-community

---

### Decision · Program_Chairs · 2024-01-22

**Decision:**

Accept (Oral)

**Comment:**

This paper focuses on robust graph structure refinement against adversarial attacks of GNNs. To solve the limitations of existing methods brought by narrow assumptions such as clean node features, moderate structural attacks, and the availability of external clean graphs, the authors proposed a self-guided GSR framework. The proposed framework utilizes a clean sub-graph with graph augmentation and group-training strategies to improve the robustness of GNNs. Extensive experiments are conducted to demonstrate the effectiveness of the proposed method.

 The authors have sufficiently addressed the reviewers concerns. The authors are advised to incorporate these changes in the final version.